# An evolutionary path to altered cofactor specificity in a metalloenzyme

Anna Barwinska-Sendra [1], Yuritzi M. Garcia[2], Kacper M. Sendra [1], Arnaud Baslé[1], Eilidh S. Mackenzie [1], Emma Tarrant[1], Patrick Card[1], Leandro C. Tabares[3], Cédric Bicep[1], Sun Un[3], Thomas E. Kehl-Fie [2,4,5✉] & Kevin J. Waldron [1,5✉]

Almost half of all enzymes utilize a metal cofactor. However, the features that dictate the metal utilized by metalloenzymes are poorly understood, limiting our ability to manipulate these enzymes for industrial and health-associated applications. The ubiquitous iron/manganese superoxide dismutase (SOD) family exemplifies this deficit, as the specific metal used by any family member cannot be predicted. Biochemical, structural and paramagnetic analysis of two evolutionarily related SODs with different metal specificity produced by the pathogenic bacterium *Staphylococcus aureus* identifies two positions that control metal specificity. These residues make no direct contacts with the metal-coordinating ligands but control the metal's redox properties, demonstrating that subtle architectural changes can dramatically alter metal utilization. Introducing these mutations into *S. aureus* alters the ability of the bacterium to resist superoxide stress when metal starved by the host, revealing that small changes in metal-dependent activity can drive the evolution of metalloenzymes with new cofactor specificity.

[1] Institute for Cell and Molecular Biosciences, Faculty of Medical Sciences, Newcastle University, Newcastle upon Tyne NE2 4HH, UK. [2] Department of Microbiology, University of Illinois Urbana-Champaign, Urbana, IL 61801, USA. [3] Department of Biochemistry, Biophysics and Structural Biology, Université Paris-Saclay, CEA, CNRS, Institute for Integrative Biology of the Cell (I2BC), 91198 Gif-sur-Yvette, France. [4] Carl R Woese Institute for Genomic Biology, University of Illinois Urbana-Champaign, Urbana, IL 61801, USA. [5] These authors contributed equally: Thomas E. Kehl-Fie, Kevin J. Waldron. ✉email: kehlfie@illinois.edu; kevin.waldron@ncl.ac.uk

M etalloproteins are critical to all aspects of life. They are ubiquitous, with as many as half of all enzymes requiring a metal cofactor for function[1,2]. Metalloenzymes are often highly specific for their cognate metal ion, exhibiting reduced activity with non-native metal cofactors in vitro and in vivo[3–5]. As a result, understanding the principles that govern the specificity of metal-protein interactions is relevant to nearly all aspects of biology, medicine and biotechnology. Yet the underlying biochemical principles that determine metal cofactor specificity, and the evolutionary processes that drive the optimization of enzyme active sites to utilize a specific metal cofactor remain unclear. This in turn limits our ability to predict the metal used by proteins from their sequence, to produce synthetic metalloenzymes to perform novel reactions for industrial applications, and to rationally design metalloenzyme inhibitors for medicine.

Our limited appreciation of how metalloprotein architecture dictates metal specificity is exemplified by the superoxide dismutases (SODs)[6]. SODs detoxify superoxide, a reactive oxygen radical produced by aerobic metabolism and by the mammalian immune system in response to infection[7]. The biological importance of SODs is illustrated by their presence in most organisms, including obligate anaerobes, and the independent evolution of three distinct dismutase protein families. These SOD families are defined by the cofactor they utilize as nickel SODs, copper and zinc SODs, and manganese (Mn) or iron (Fe) SODs[6]. Mn/Fe-dependent SODs are the most common, and historically were thought to be strictly either manganese- or iron-specific. We recently established that a Mn/Fe SOD from the bacterium *Staphylococcus aureus* exhibits equal activity with either manganese or iron. These versatile enzymes are termed cambialistic SODs (camSOD)[8]. In addition to the camSOD (SodM), *S. aureus* also possesses a second, manganese-dependent SOD (SodA)[8–10]. Although cambialistic SODs had previously been described[6,11–15], their biological importance was questioned. However, the *S. aureus* camSOD contributes to infection by enabling the bacterium to maintain a defense against superoxide when manganese starved by the host[8,16,17]. All members of the Mn/Fe SOD family are related in sequence, exhibit identical protein folds, and

coordinate their metal ion using identical ligands[6], making it unclear why some enzymes absolutely require manganese for catalysis (MnSOD), while others require iron (FeSOD), and still others show metal cofactor flexibility (camSOD).

The metal used by a protein is not permanently fixed, and can change in response to environmental pressures[2,18]. For example, iron was readily soluble in the anaerobic oceans during life's early evolution and early organisms are thus thought to have been iron-philic[18,19]. However, oxygenation by early photosynthetic organisms reduced the availability of iron[18,19]. The resulting biological iron deficiency would have imposed selective pressure to adapt iron-dependent enzymes to use non-iron cofactors[18–20]. While supported by bioinformatic analyses[18,21], no experimental evidence has been presented demonstrating the evolutionary process by which a change in metal specificity has evolved through iterative mutation[20].

Here, we exploit the close relationship between the staphylococcal SODs to understand how evolutionary changes in metal utilization occur. Genomic analysis shows the camSOD likely evolved from a manganese-specific predecessor that subsequently underwent neofunctionalization, a defined evolutionary process in which mutations rapidly accumulated in the duplicated gene during a period of functional redundancy, resulting in gain of a new beneficial function[22]. Integrated structural, biochemical, and electron paramagnetic resonance (EPR) studies reveal that two such mutations have altered amino acid residues in close spatial proximity to the SOD active site, driving the change in camSOD metal specificity. When these residues are reciprocally swapped, the metal specificities of the MnSOD and camSOD are largely interconverted. Remarkably, these residues possess non-polar sidechains located in the metal's secondary coordination sphere, and make no direct contacts to the metal-coordinating ligands. These subtle changes regulate the electronic structure and redox properties of the catalytic metal ion, dictating which metals the enzymes can use. Leveraging these findings reveals that small increases in iron-dependent catalysis by camSOD enhance the ability of *S. aureus* to overcome the immune response. Collectively, our data show how subtle changes to metalloenzyme architecture can dramatically alter the metal ion's reactivity and drive the evolution of isozymes with new cofactor specificity.

## Results

**The two *S. aureus* SODs exhibit extensive similarity.** Initially, we comprehensively characterized which metals the MnSOD and camSOD could use. The MnSOD was active only with manganese, while the camSOD was active with manganese or iron (Supplementary Fig. 1 and Table 1). As with other characterized cambialistic SODs[12,14,23], the activity of *S. aureus* camSOD is significantly lower than that of the MnSOD, suggesting that cambialism represents an evolutionary compromise, where catalytic activity is sacrificed at the expense of enabling cofactor flexibility. Alignment of the *S. aureus* SOD amino acid sequences demonstrated that the two proteins share 75% identity (Supplementary Fig. 2). Next, we examined the *S. aureus* SODs for structural differences using circular dichroism (CD) spectroscopy. The iron- and manganese-loaded forms of both proteins showed very similar CD spectra (Fig. 1a), consistent with both SODs containing comparable secondary structure composition in solution (Supplementary Table 1). We then determined three dimensional crystal structures of the iron-loaded forms of each SOD by X-ray diffraction and compared them with structures[24] of the manganese-loaded counterparts (Supplementary Table 2). All four forms adopted near-identical architectures (Fig. 1b and Supplementary Fig. 3) whose polypeptide backbones could be overlaid with only minor deviations (Supplementary Table 3).

**Table 1 Enzymatic activity of all SOD forms from this study.**

| Enzyme | Mutation | Mn activity[a] | Fe activity[a] | Cambialism ratio[b] |
|---|---|---|---|---|
| MnSOD | Wild type | 1836 ± 79 | 4 ± 1 | 0.002 |
| | F19I | 1734 ± 61 | 5 ± 1 | 0.003 |
| | G159L | 942 ± 69 | 39 ± 4 | 0.041 |
| | L160F[c] | 1167 ± 92 | 19 ± 13 | 0.016 |
| | G159L-L160F | 173 ± 21 | 84 ± 12 | 0.486 |
| | F19I-G159L-L160F | 526 ± 49 | 57 ± 7 | 0.108 |
| camSOD | Wild type | 251 ± 5 | 250 ± 10 | 0.996 |
| | I19F | 216 ± 31 | 325 ± 1 | 1.505 |
| | L159G | 808 ± 17 | 24 ± 3 | 0.030 |
| | F160L[c] | 342 ± 5 | 111 ± 20 | 0.325 |
| | L159G-F160L | 687 ± 33 | 11 ± 1 | 0.016 |
| | I19F-L159G-F160L | 789 ± 54 | 39 ± 2 | 0.049 |

Enzymatic activities of all variants of *S. aureus* MnSOD and camSOD, in each of their metal-loaded forms, were assayed using a commercial SOD Activity Assay kit (Sigma). Each enzyme was assayed in triplicate using independent biological replicates, and error values quoted represent the standard deviation (SD) from the mean value given. All samples were assayed n = 3 except for three forms of the triple mutant variants that were assayed n = 4. Source data are provided as a source data file.
[a]Values represent Units mg$^{-1}$ protein ± SD.
[b]The cambialism ratio (CR) is calculated by dividing the iron-dependent activity by the manganese-dependent activity, with a value of 1 representing a perfectly cambialistic enzyme. Enzymes that are cambialistic are defined herein as having a CR in the range 0.1–10.00.
[c]Whereas all other samples were assayed simultaneously to circumvent noticeable day-to-day variability in the absolute activity detected using the commercial assay, the single-mutant variants with mutations at position 160 were assayed separately. To ensure comparability between the datasets, these variants were assayed simultaneously alongside samples of all wild-type forms, and small discrepancies in the absolute values were normalized by scaling using those wild-type activities as controls.

Crucially, the metal-binding ligands adopted near-identical spatial positions in all four structures (Fig. 1c), with insignificant changes to metal-ligand bond lengths or angles within the crystallographic resolution (Supplementary Tables 4 and 5). This is consistent with our previous EPR study that showed the positions of the protons on the metal-coordinated solvent and histidine ligands are identical[24].

**Identification of residues proximal to the active site**. The structural similarity of the two SODs suggested that subtle differences must explain their different metal specificity. By overlaying the crystal structures of MnSOD and camSOD, all 50 amino acids that differed between the two proteins were spatially

localized (Fig. 2a). Few of these were at the dimerization interface (Supplementary Fig. 4), consistent with this region being highly conserved in SODs (Fig. 2b), and most were surface localized (Fig. 2c). However, we identified three residues (19, 159, and 160) internal to the protein that varied between MnSOD and camSOD that were spatially close (<10 Å) to the metal (Fig. 2d), where non-polar sidechains (Leu or Ile for Phe, or Gly for Leu) were swapped. Notably, these sidechains in the secondary coordination sphere made no direct contacts with the metal-coordinating sidechains in the primary coordination sphere.

**Metal-specificity is determined by sequence positions 159/160**. To study their role in metal specificity, we prepared variant forms

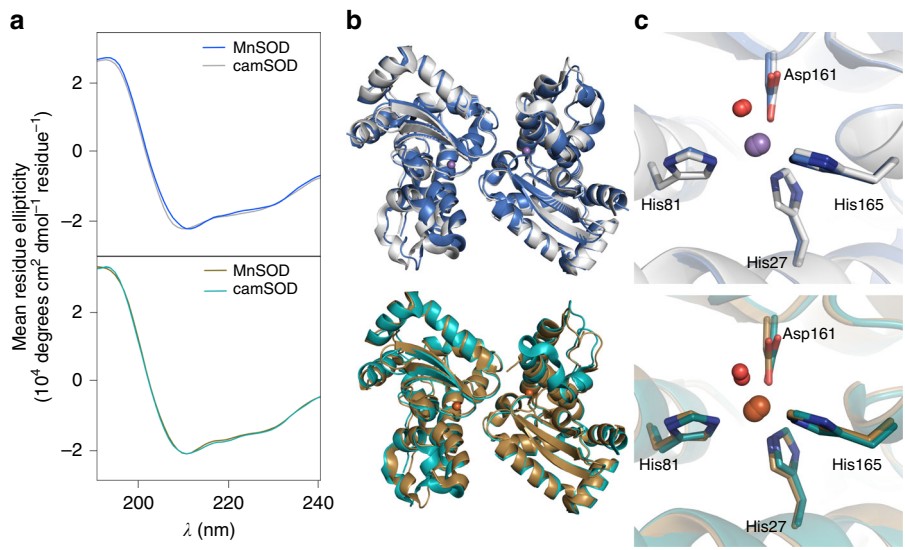

**Fig. 1 Extensive structural similarity of *S. aureus* MnSOD and camSOD. a** Overlaid CD spectra ($n = 1$) of the (upper panel) manganese-loaded (MnSOD in blue; camSOD in gray) and (lower panel) iron-loaded (MnSOD in gold; camSOD in teal) isoforms, demonstrating similar secondary structure content of the two isozymes in solution. **b** Superimposed crystal structure cartoons of (upper panel) manganese-loaded forms[24] and (lower panel) iron-loaded forms of each enzyme, with the proteins colored as per panel (**a**), and with manganese ions, iron ions, and waters shown as purple, orange, and red spheres, respectively. Analysis of all four structures show that the polypeptide backbones of all isoforms of the staphylococcal SODs are essentially identical to within the ~2 Å resolution of the structural data (Supplementary Tables 3–5). **c** No significant differences within the structural resolution were detectable in the metal coordination environment that could explain the disparate metal specificity of their catalysis.

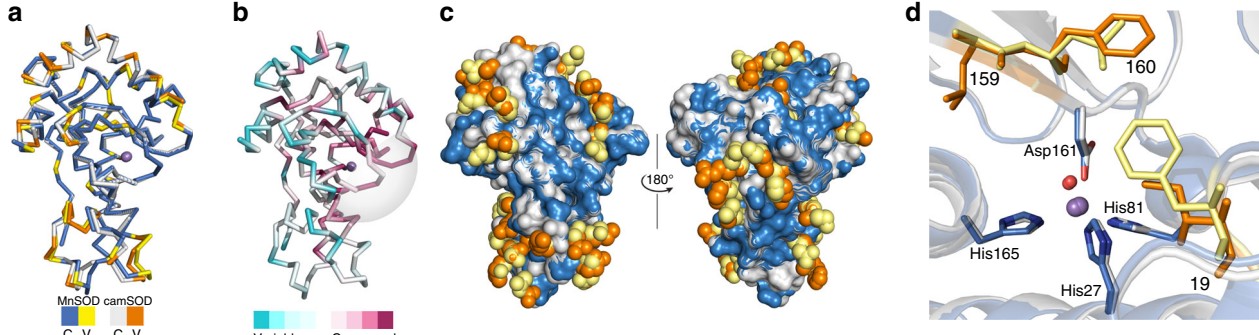

**Fig. 2 Mutation of two residues located close to the active site inverts specificity. a** Superimposed ribbon representation structures of the polypeptide backbones showing conserved (C) regions of the manganese-loaded forms of MnSOD (blue) and camSOD (gray), with all variable (V) residues highlighted (yellow and orange, respectively). **b** Analogous ribbon representation of the structure of *S. aureus* MnSOD, illustrating the localization of variable (cyan) and conserved (purple) regions of SOD sequence based on ConSurf analysis[59] of 500 SOD sequences (calculated conservation scores represented by the color spectrum key). Approximate dimerization interface is indicated (gray). **c** Superimposed space-filling models of MnSOD and camSOD, colored as in panel (**a**), illustrating that >85% of sidechains that vary between the *S. aureus* SODs are surface-exposed (2.5 Å$^2$ surface exposure cutoff used). **d** Magnified view of the superimposed cartoon representations of MnSOD and camSOD structures, colored as in panel (**a**), showing the spatial location of the three residues targeted for mutagenesis in proximity to the manganese ion (purple).

of MnSOD and camSOD in which the amino acids at the three candidate positions were reciprocally swapped. The recognition that SOD metal specificity in vitro and in vivo is better described as a spectrum than as binary possibilities highlights the need for a quantitative measure of cofactor plasticity to be used alongside absolute activity. Herein, we define the term cambialism ratio (CR) as such a measure, being the ratio of iron-dependent activity to manganese-dependent activity detectable using a spectro-photometric assay[8]; this value is close to zero for MnSODs (CR = 0.002 for *S. aureus* MnSOD), close to unity for camSODs (CR = 0.996 for *S. aureus* camSOD), and has large values for FeSODs (CR > 800 for *E. coli* FeSOD[25]).

Reciprocal mutations introduced at position 19 had negligible effects on the metal specificity of either enzyme (Table 1). Conversely, exchanging the residues at position 159 significantly impacted metal specificity. In the MnSOD, conversion of Gly159 to Leu (MnSOD Gly159Leu) increased its Fe-dependent activity ~10-fold relative to wild-type MnSOD while decreasing its manganese-dependent activity ~2-fold (Table 1). While remaining a highly manganese-specific enzyme, this variant exhibited slightly increased cambialism (CR = 0.048). The reciprocal mutation in camSOD, converting Leu159 to Gly (camSOD Leu159Gly), diminished its iron-dependent activity >10-fold relative to wild-type camSOD while increasing its manganese-dependent activity >3-fold, resulting in an enzyme variant with a MnSOD-like activity profile (CR = 0.030) (Table 1). These results indicate that the residue at position 159 has a major effect on the ability of the *S. aureus* SODs to use iron and manganese, but the contribution of this residue alone was insufficient to fully explain the disparate metal specificities of the two *S. aureus* enzymes.

Interconversion of the residue at position 160 alone also influenced metal specificity (Table 1). The MnSOD variant (Leu160Phe) was similar to the MnSOD Gly159Leu variant, exhibiting diminished manganese-dependent activity and increased iron-dependent activity (CR = 0.016) but to a lesser extent than in Gly159Leu, whereas the camSOD variant (Phe160Leu) showed slightly modified activity with both metals but remained largely cambialistic (CR = 0.325). Crucially, combining this mutation with that at position 159 further inverted the enzymes' metal specificity. The MnSOD double mutant (Gly159Leu-Leu160Phe) showed a substantial difference in metal utilization from that of the wild type or the single-mutant variants. The iron-dependent activity of the MnSOD double mutant was increased >20-fold relative to wild-type MnSOD, resulting in a highly cambialistic enzyme that displayed activity with manganese and with iron that differed only ~2-fold (CR = 0.486). It's notable that, like in wild-type camSOD, this highly cambialistic variant exhibited diminished activity with manganese, illustrating that gaining cofactor flexibility requires a compromise with catalytic efficiency. It further demonstrates the importance of comparing both the absolute activity and the cambialism ratio when assessing metal specificity. Conversely, the camSOD double mutant (Leu159Gly-Phe160Leu) had greater activity with manganese and reduced activity with iron relative to the wild type or the single mutant (camSOD Leu159Gly). This shift in camSOD metal preference resulted in a highly manganese-specific enzyme (CR = 0.016) (Table 1). Cumulatively, these results demonstrate that just two mutations in the secondary coordination sphere largely interconvert the metal specificity of camSOD and MnSOD.

Next, triple mutant variants, in which residues 19, 159 and 160 were all swapped, were evaluated. Unexpectedly, the MnSOD triple mutant (Phe19Ile-Gly159Leu-Leu160Phe) exhibited greater activity with manganese and less activity with iron (CR = 0.108) than the double mutant (Table 1). The camSOD triple mutant (Ile19Phe-Leu159Gly-Phe160Leu) had increased activity

with both metals relative to the double mutant (camSOD Leu159Gly-Phe160Leu). These studies demonstrate that positions 159 and 160 are important in regulating the reactivity of the metal cofactor. Strikingly, mutations at these positions produced profound effects on metal specificity, yet did so with essentially no changes to the protein backbone structure, at least within the resolution of the crystal structures (Supplementary Fig. 5). The physical structures of the metals' primary ligand spheres were the same, as were the substrate access channels and hydrogen-bonding networks that are important for proton-coupled electron transfer[6].

**Metal utilization correlates with the metal's electronic properties.** Despite the similarity of the crystal structures, the mutations did have direct effects on the electronic structure and electrochemical properties of manganese centers. High-field electron paramagnetic resonance (HFEPR) spectra of the Mn(II)-loaded *S. aureus* SOD wild-type forms[24] and the six variants were distinct (Fig. 3a). These spectra are determined by the $D$ and $E$ values of the zero-field interactions (ZFI) of the Mn(II) centers (Supplementary Fig. 6, Supplementary Eqs. 1–6, and Supplementary Note 1). Since the ZFI arises from the spin-orbit and magnetic interactions of the five unpaired Mn(II) electrons, this unambiguously demonstrated that the electronic structures of the metal centers were different. The sum, $|D|+E$, which measures the size of the Mn(II) ZFI, has been shown to correlate semi-quantitatively with manganese-dependent activity, with cambia-listic SODs exhibiting intermediate values between the low values of Mn(II)-loaded FeSODs and the high values of MnSODs[26]. As reported previously, the wild-type *S. aureus* proteins were consistent with this trend[24], but the current measurements shed greater light. The values of $|D| + E$ and the manganese-dependent enzymatic activities of seven of the enzyme forms (Supplementary Table 6) exhibited a significant linear correlation (Supplementary Fig. 7). The furthest departure from this correlation was camSOD Leu159Gly-Phe160Leu. This exception notwithstanding, the significance of this correlation can be interpreted from the results of studies on Mn(II/III)- and Fe(II/III)-containing, 4′-substituted 2,2′:6′,2″-terpyridine complexes. These compounds are structurally homologous and exhibit metal-specific or cambialistic SOD-like activity depending on the electron-donating/withdrawing strength of the 4' substituent, quantified by its Hammett $\sigma$ values[27]. The reduction potentials of both the manganese and iron complexes exhibit a linear dependence on $\sigma$, as do the ZFI of the Mn(II) complexes[26]. Hence, the relationship between metal-specific activity, redox tuning through indirect secondary coordination sphere electronic effects, and Mn(II) ZFI in the enzymes is entirely consistent with the effects of subtle charge polarization in the simpler, better understood chemical systems.

Further evidence that the secondary coordination sphere residues influence the redox properties of the manganese cofactor comes from observations of their auto-oxidation properties. Mn(III) weakly absorbs in the 400–600 nm region[25,28,29]. The spectra of the two wild-type SODs in their resting state (equilibrated with ambient oxygen) had similar shapes, but the MnSOD peak intensity was greater (Fig. 3b). This demonstrated the MnSOD was more susceptible to auto-oxidation under these conditions than the camSOD, whereas the converse was true for the double mutants (Fig. 3b). These trends in auto-oxidation correlated with manganese-dependent enzyme activity (Table 1). One half reaction catalyzed by SODs is the oxidation of superoxide to molecular oxygen, and auto-oxidation is formally the reverse of this reaction (Supplementary Eq. 7). If auto-oxidation proceeds by the reverse mechanism (Supplementary Eqs. 8–12), then any changes in metal reduction potential will affect this reaction in

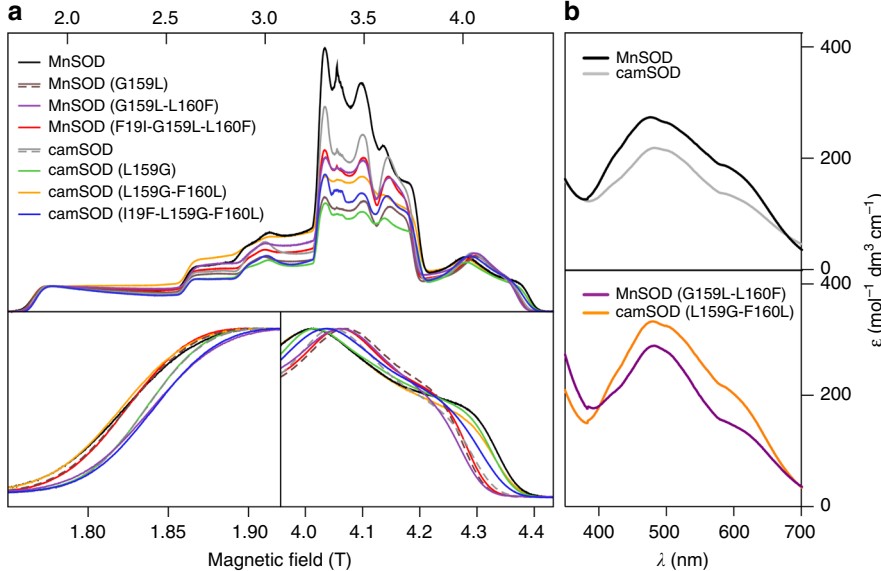

**Fig. 3 Differences in electronic structure and auto-oxidation of the SODs. a** The 94 GHz Mn(II) field-swept echo HFEPR spectra (upper panel) of each of the dithionite-reduced, manganese-loaded forms of the *S. aureus* wild type and variant SODs (labeled in the color key, top left), acquired at 5–6 K ($n = 1$). The lower panels show expanded views of the (left) lower and (right) upper field shoulders, respectively, which are indicated by arrows in the upper panel. SOD mutations gave rise to detectable differences in the enzymes' HFEPR spectra and their ZFI (Supplementary Figs. 6–7 and Supplementary Table 6), demonstrating altered electronic structure and redox properties caused by the mutations. **b** UV/visible absorption spectra of the manganese-loaded forms of the (upper) wild type and (lower) double mutant variants of MnSOD (black and purple, respectively) and camSOD (gray and orange, respectively). MnSOD exhibits greater auto-oxidation at rest than camSOD, illustrated by increased absorbance of Mn(III) at ~480 nm, a trend which is reversed in the double mutant variants that also exhibited reciprocal changes in metal specificity (Table 1). All samples (800 µM) were equilibrated with ambient aerobic conditions in 100 mM phosphate buffer, pH 7.5, 100 mM KCl, 1 mM EDTA before spectra were acquired. Spectra were collected on independent biological replicates ($n = 2$), mathematically converted to extinction coefficient ($\varepsilon$) using the Beer-Lambert law, and a representative spectrum is shown for each variant.

both directions (Supplementary Note 2). There are other factors that could differentially affect the two reactions, namely substrate accessibility and proton availability. However, within the limits of the resolution of the crystal structures, we found no evidence of changes in structure that were likely to affect superoxide/ dioxygen motion to/from the metal ion or proton availability, with the positions of the residues around the active site remaining unchanged (Supplementary Fig. 5). While our structural analysis suggested these alternative possibilities were unlikely, we also measured the susceptibility of the oxidized form of the manganese-loaded SODs to reduction by absorption spectroscopy. Each enzyme was oxidized with permanganate, and then titrated with increasing concentrations of dithionite while monitoring the disappearance of the oxidized Mn(III) species using UV/visible detection (Supplementary Fig. 8). The camSOD was fully reduced at lower concentrations of dithionite than the MnSOD, consistent with the proposed difference in reduction potentials between the wild-type isozymes (Supplementary Eqs. 13–17). Importantly, this trend was also reversed in the double mutant variants, with MnSOD G159L-L160F becoming reduced at lower concentrations of dithionite than camSOD L159G-F160L. The SOD metal site is only accessible via a narrow solvent channel, making small redox-active molecules inefficient electrochemical mediators for SODs, thus precluding direct reduction potential measurements[30].

Altogether, the spectroscopic data indicate that the two wild-type *S. aureus* SODs and the two double mutants differ in their manganese redox properties. The HFEPR spectra demonstrated that the electronic structure of their Mn(II) centers were significantly different. The auto-oxidation data and dithionite reduction data both suggest that the two wild-type SODs have different reduction potentials. These differences are inverted in

the double mutants in both assays, which correlates with the inversion of manganese-dependent activity in these variants (Table 1). We conclude therefore that the altered metal specificity caused by the mutations is related to changes in reduction potential of the metal ions. The double mutation in MnSOD resulted in an altered electronic structure and re-tuning of the manganese reduction potential to more closely match that of camSOD and more catalytically active with iron, whereas the camSOD mutations altered reduction potential to make it more similar to MnSOD and more catalytically active with manganese.

**S. aureus camSOD likely arose after a duplication of MnSOD.** A challenge in studying metalloenzyme neofunctionalization is a lack of knowledge regarding the properties of the predecessor from which it evolved. To evaluate the evolutionary relationship between the *S. aureus* SODs, we performed a maximum likelihood phylogenetic analysis using 2691 aligned Mn/Fe SOD sequences from bacteria, archaea and eukaryotes (Supplementary Data 1). The overall topology of the resulting phylogenetic tree was consistent with that derived from a previous analysis of a more limited sequence library[19] (Fig. 4a). Combining amino acid correlation analysis (Supplementary Data 2) with the phylogenetics (Supplementary Fig. 9) showed that the analyzed SOD sequences could be grouped into two distinct subtypes based on the presence of highly conserved GGH or AAQ motifs, as well as a third, diverse group lacking either motif (Fig. 4a, b and Supplementary Fig. 10). We annotated known SOD metal specificities onto the tree, including only examples where activity with both manganese and iron cofactors were definitively tested, which demonstrated that the GGH subgroup contains predominantly manganese-dependent SODs[14,31] and the AAQ subgroup contains primarily iron-dependent SODs[23,25].

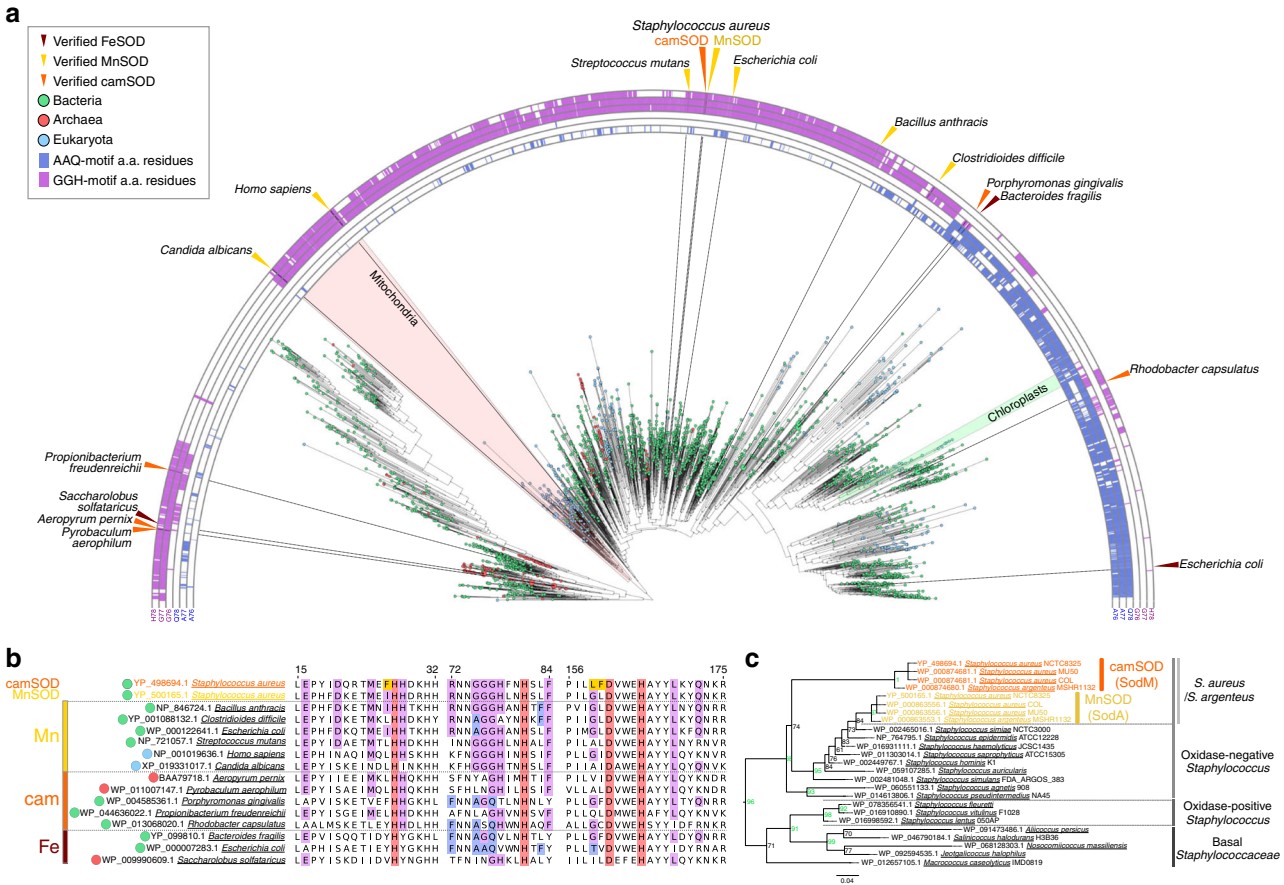

**Fig. 4 *S. aureus* camSOD evolved from a manganese-specific predecessor. a** Amino acid correlation (defined in the text as the AAQ group, blue, and the GGH group, magenta) and unrooted maximum likelihood phylogenetic tree based on alignment of amino acid (a.a.) sequences of 2691 Mn/Fe SOD homologs from bacteria (green circles), eukaryotes (sky blue circles), and archaea (red circles). Known metal-specificities of characterized enzymes (MnSOD = yellow, FeSOD = brown; camSOD = orange) are annotated with triangles. **b** Fragment of the alignment of selected SOD sequences of known metal specificity, with the metal-binding residues (red) and three residues in *S. aureus* camSOD (yellow) that were targeted for mutagenesis shown. Groups 1 and 2 of coevolving residues, identified using amino acid correlation analysis, were mapped onto the alignment, illustrating the GGH or AAQ motifs and associated correlated amino acids. The correlated residues, biological origin and metal specificity are indicated using the color scheme described in **a**, and sequence numbering is based on the *S. aureus* SODs. **c** Sub-tree extracted from the SOD tree in panel **a** containing all identified staphylococcal SOD sequences. Support values correspond to maximum likelihood bootstrap values from 100 rapid bootstrap replicates, with values >90 shown in green. The scale bar indicates number of substitutions per site. *Macrococcus* and the oxidase-positive staphylococci form an out-group to a strongly supported grouping of oxidase-negative staphylococci, including *S. aureus*, consistent with published Staphylococcaceae phylogenies[19,40,41]. Within the oxidase-negative staphylococci, all MnSOD (yellow) grouped together while camSOD homologs (orange) formed an out-group to the MnSODs.

Nonetheless, confirmed cambialistic SODs were present in both subgroups[15,32] and a GGH-containing enzyme has been shown to be an FeSOD[33] (Fig. 4a, b), implying that the three metal specificities can and do interchange evolutionarily.

Inspection of the staphylococcal SOD protein similarity network and the tree (Supplementary Fig. 11) demonstrated that MnSOD is present in all staphylococcal genomes, whereas camSOD was identified only in the genomes of *S. aureus*, and of *S. argenteus* and *S. schweitzeri*, both recognized as early-branching lineages of the *S. aureus* tree[34,35]. As MnSOD is the closest homolog to camSOD, the most parsimonious explanations are that the *sodM* gene that encodes camSOD arose from a duplication of *sodA*, encoding MnSOD, or that *sodM* was acquired by horizontal gene transfer from a close relative. Regardless, either model predicts that the newly acquired *sodM* would have originally encoded a manganese-specific enzyme. Increased branch length indicates an increased rate of evolutionary change for the ancestral SodM, while SodA remained under purifying selection[22] (Fig. 4c). The camSOD sequence is highly conserved across all *S. aureus* (99% identity), and this homology extends to the second SOD found in *S. argenteus* (97.5%) and *S. schweitzeri* (99%) (Supplementary Fig. 2). Notably Leu159 and Phe160 in camSOD are strictly conserved between *S. aureus*, *S. schweitzeri* and *S. argenteus*. Thus, we conclude that the acquired *sodM* gene originally encoded a MnSOD, which subsequently evolved into the modern camSOD during the emergence of *S. aureus*.

**Acquisition of camSOD coincided with virulence factor expansion.** To understand the wider context of camSOD emergence within the *S. aureus* lineage, we performed whole genome comparisons and a protein similarity network analysis (Supplementary Data 3) of genomes sampled across the staphylococcal phylogenetic tree (Supplementary Data 4). The *sodA* gene encoding MnSOD was located within a genomic region rich in essential genes[36–38] (Supplementary Data 5) that are highly conserved across the staphylococci (Fig. 5a, b). Conversely, the *sodM* gene encoding camSOD is located within a highly variable genomic region close to the chromosomal origin that is enriched in virulence factors[39] (Supplementary Data 5), consistent with

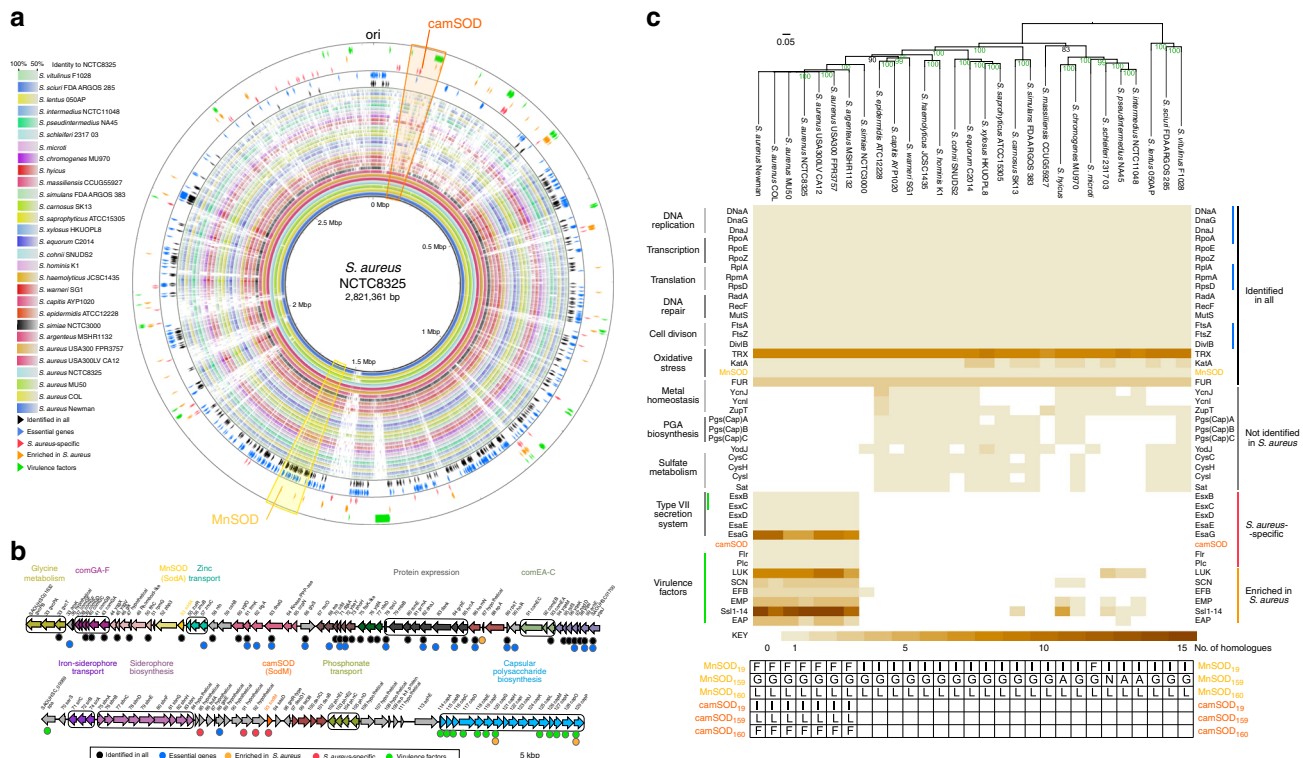

**Fig. 5 Evolution of camSOD coincided with expansion of virulence factor network. a** Circular genome alignment comparing 29 analyzed staphylococcal genomes, using *S. aureus* strain NCTC8325 as reference genome. The *sodA* gene (yellow), encoding MnSOD, is located in a genomic region enriched in essential genes (blue) that is highly conserved across the staphylococci, whereas the *sodM* gene (orange), encoding camSOD, is located in a variable genomic region enriched in virulence factors (green). **b** Expanded view of the local genomic context of the *sodA* (upper panel) and *sodM* (lower panel) genes. **c** Heatmap depicting the results from analysis of protein similarity networks, which identified protein families whose copy numbers (see color key) have been altered during evolution of the staphylococci (see phylogenetic tree in the upper panel). Protein families are represented on horizontal lines, and are grouped as Identified in all (black), Not identified in *S. aureus* (gray), *S. aureus*-specific (red), Enriched in *S. aureus* (orange) as described in the Methods. Genes inferred to be essential in analyzed *S. aureus* genomes are illustrated by blue bars. The lower panel illustrates the amino acid residues present at positions 19, 159, and 160 of each SOD isozyme present in the respective genome.

these factors having been primarily acquired via horizontal transfer of mobile elements[40,41]. We screened the protein similarity networks to identify proteins with distributions similar to that of camSOD (Supplementary Data 3); proteins present in all *S. aureus*/*S. argenteus* but absent in other staphylococci, likely to have been present in their last common ancestor. We observed that the genomes that encoded camSOD possessed an expanded set of proteins involved in microbial competition, host interaction, immune evasion and virulence[40,42] (Fig. 5c and Supplementary Data 3 and 5). Some of these were unique to *S. aureus*/*S. argenteus*, whereas others were present in other staphylococci but were enriched within this clade, and included proteins involved in capsule biosynthesis (Cap), immunomodulation (Flr, Efb, Emp), secreted effectors (Esx) and toxins (Luk). A component of the Isd heme-uptake system (IsdD) was also unique to this clade (Fig. 5c). Indeed, we observed that *S. aureus* genomes are enriched in components that function in iron uptake via heme or siderophores[43] (Fig. 5c, Supplementary Fig. 12, and Supplementary Data 6). We conclude that the evolution of camSOD in the last common ancestor of *S. aureus*/*S. argenteus* coincided with the expansion of their repertoire of virulence genes[40,41], and that its genomic context are consistent with its established role in pathogenicity[8].

**A single mutation alters resistance to host-derived stresses.** The camSOD is a virulence factor that enables *S. aureus* to resist oxidative stress during infection when starved of manganese by

the host[8]. Our analysis suggested that camSOD likely evolved from a manganese-specific predecessor, implying that evolution subsequently adapted the manganese-specific precursor, resulting in the extant cambialistic enzyme. We thus aimed to leverage our ability to produce camSOD enzymes with altered metal specificity to test whether the camSOD's iron-dependent catalysis directly contributes to *S. aureus* overcoming the host immune response, and if small changes in iron-dependent activity are sufficient to provide a functional advantage in the presence of natural stressors.

A single mutation was introduced into the *S. aureus* chromosome, converting the wild-type camSOD to its Leu159Gly variant, which shows reduced ability to utilize iron (Table 1). This strain was evaluated, alongside wild-type *S. aureus* and a Δ*sodM* mutant that lacks camSOD entirely, for its ability to resist conditions representative of dual host-imposed stresses, manganese starvation and oxidative stress. Manganese starvation was imposed using the manganese-binding immune effector calprotectin, which can accumulate at sites of infection[8,16,17] at concentrations in excess of 1 mg ml⁻¹, while superoxide was generated by addition of paraquat[8,17]. Consistent with previous studies[8,17], the strain lacking camSOD was more sensitive to these stresses than wild type (Fig. 6a). The strain carrying the Leu159Gly variant camSOD enzyme was also more sensitive to these stresses (Fig. 6a), and its SOD activity showed altered susceptibility to inhibition by peroxide (Fig. 6b, c), consistent with altered metal loading. These results demonstrate that the

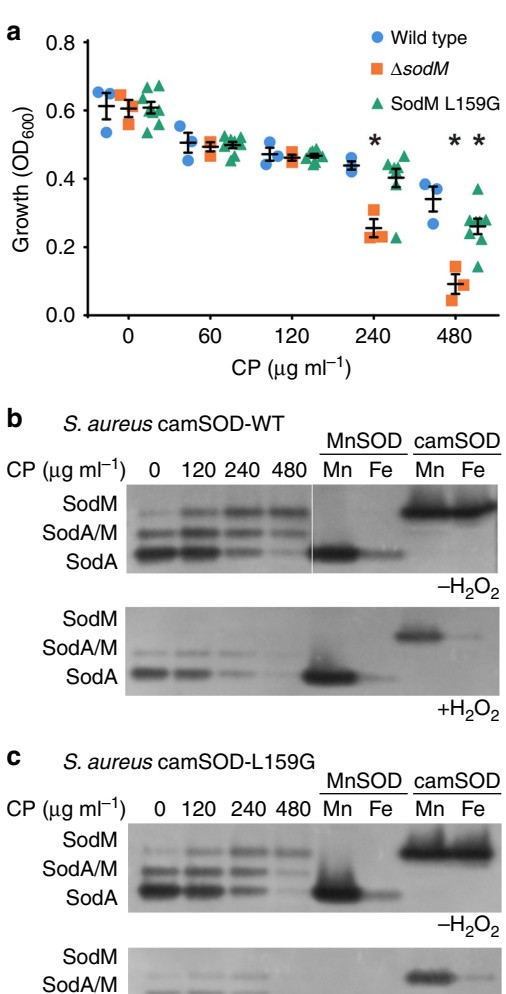

**Fig. 6 Iron-dependent camSOD activity enables *S. aureus* to resist host stresses. a** The Δ*sodM* mutant strain (lacking the gene encoding camSOD-orange squares) of *S. aureus* shows greatly diminished growth in the presence of paraquat (PQ) relative to wild type (WT-blue circles) under manganese-depleted conditions imposed by the presence of the human protein complex calprotectin (CP)[8,16,17]. *S. aureus* cells expressing the Leu159Gly variant of camSOD from the native *sodM* locus (green triangles) also show reduced growth relative to the wild type. Note that the asterisk represents $p < 0.05$ via two-way ANOVA with Tukey's post-test performed in Graphpad Prism when compared to wild-type bacteria grown in the same concentration of calprotectin. Multiple independent clones of the Leu159Gly variant ($n = 8$) were assayed at the same time as those of the wild type ($n = 3$) and the Δ*sodM* mutant strain ($n = 3$). Each data point represents an independent biological replicate. Black bars represent the mean, with error bars showing standard error of the mean (SEM). The diminished ability of the camSOD variant to use iron (Table 1) renders *S. aureus* less capable of overcoming the host immune response. **b**, **c** In-gel SOD activity assays using extracts (5.4 μg) prepared from **b** wild type *S. aureus* and **c** the strain expressing the Leu159Gly camSOD variant, cultured under identical conditions to those used in **a**, demonstrate that both forms of the SodM enzyme are detectable at similar levels (upper panels). Treatment of the extracts (left) or recombinant enzymes (right) with $H_2O_2$ (lower panels), a specific inhibitor of the iron-loaded form[8], demonstrated that the camSOD was loaded with iron in wild-type cells. The MnSOD enzyme is labeled as SodA, camSOD as SodM. This is a representative gel from a triplicate analysis of independent biological replicates ($n = 3$). Note that molecular weight markers were not used in these native gels, with SOD bands identified in the cell extracts by comparison of their mobility with the purified recombinant proteins. Source data for all panels are provided as a source data file.

ability of camSOD to perform iron-dependent catalysis enables *S. aureus* to resist oxidative stress during physiological manganese deprivation.

Collectively, our data are consistent with a model in which the evolution of camSOD was driven by interaction of the ancestor of *S. aureus* with the host immune system (Fig. 7). Acquisition of a redundant gene encoding a second manganese-specific SOD was followed by a period of rapid evolution of the protein sequence, which conferred the ability to utilize iron for catalysis. This period coincided with the expansion of the organism's repertoire of virulence-associated proteins, enabling *S. aureus* to survive immune assault and establish infection better than less pathogenic relatives[40]. We thus propose that the selection pressure that drove the neofunctionalization of camSOD from a manganese-specific precursor after acquisition involved the combined immune-mediated stresses of the oxidative burst and manganese starvation[7,8,16,17].

## Discussion

Metalloproteins are ubiquitous in biology, thus understanding the principles that govern metal cofactor utilization has implications for diverse aspects of biology, medicine, and biotechnology. However, the molecular factors that dictate the metal cofactor utilized by metalloenzymes, and how iterative changes to the protein sequence over time result in a change in metal specificity, remain poorly understood. Our inability to accurately predict the metal utilized by the Mn/Fe SODs, a widely distributed family of metalloenzymes, exemplifies these deficiencies. In this study, we developed the pair of *S. aureus* SODs as a powerful model system with which to interrogate how metal specificity can change over evolutionary time, enabling future studies to determine how the architecture of a metalloenzyme imposes cofactor specificity. Exploiting this platform demonstrated that evolution can effect substantial changes in metal specificity by making modest changes in the chemical nature of secondary coordination sphere residues that regulate metal reactivity.

Evolutionary distance between SODs with differing metal specificity has limited our ability to elucidate which residues dictate metal utilization[6,12,26,29]. Leveraging the extensive structural and sequence homology between the staphylococcal SODs identified two key residues that dictate metal-specificity. Remarkably, the chemical differences between the sidechains of these residues are minor, with Gly (MnSOD) and Leu (camSOD) at position 159, and Leu (MnSOD) and Phe (camSOD) at position 160. While both residues are close to the active site, neither sidechain makes direct contact with the metal or its coordinating ligands, nor do they affect the overall architecture of the active site. Despite this, HFEPR and UV/visible spectroscopy revealed that interconversion of these residues profoundly alters the electronic structure and redox properties of the manganese relative to the wild-type enzymes[24]. These data confirmed that MnSOD is more susceptible to auto-oxidation than camSOD, and crucially, this difference was inverted in the variants that also exhibited inverted metal specificities. These results support a model in which the SOD architecture selectively tunes the cofactor's reduction potential[6,28,44], and identify positions 159 and 160 as playing a critical role in this electronic modulation. Broadly, these results highlight the important contribution of secondary coordination sphere residues in determining metal specificity. Further, they illustrate that even minor secondary sphere changes can have profound impacts on metal-specificity, which is relevant to the study of natural metalloproteins, and the de novo design of synthetic metalloenzymes[45–49].

The residue equivalent to Gly159 in *S. aureus* MnSOD was reported as being generally Gly in both manganese-dependent

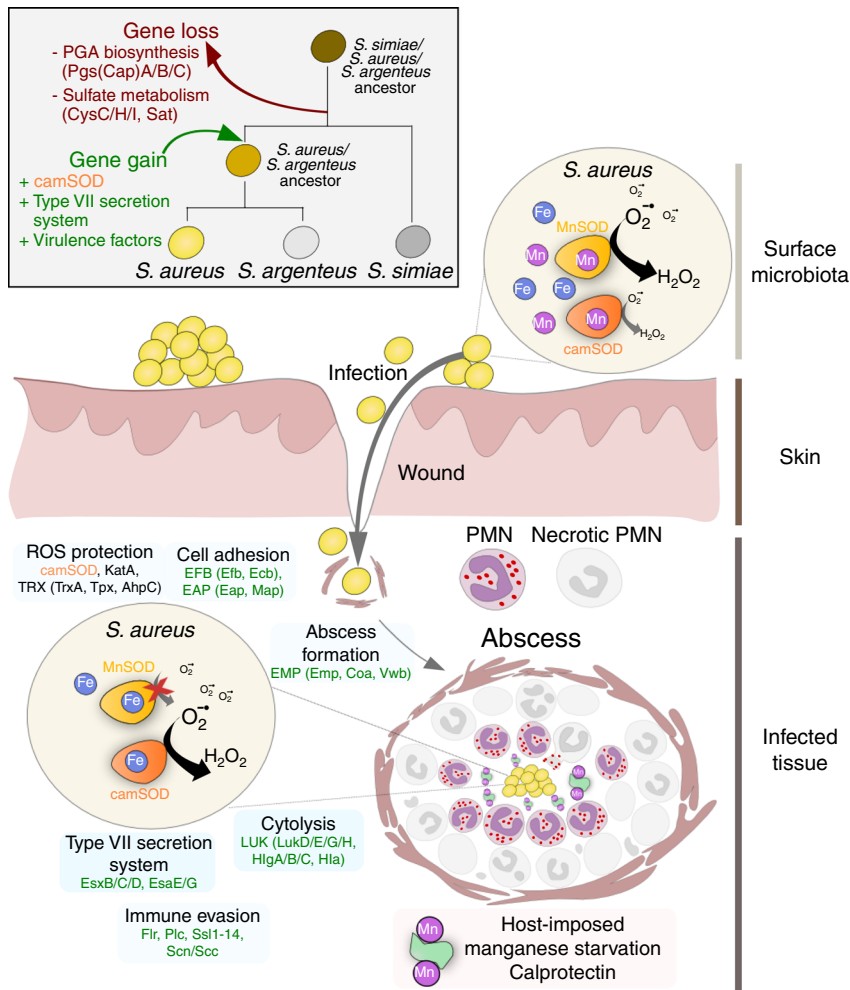

**Fig. 7 Evolution of camSOD was driven by altered metal availability in the host.** Schematic model depicting the evolution of *S. aureus* camSOD (orange) under selection pressure caused by exposure of the *S. aureus* ancestor to host-imposed manganese-starvation mediated by calprotectin. Acquisition and evolution of camSOD enabled the ancestor of *S. aureus* to maintain its antioxidant defense under these conditions using its iron-dependent catalysis, and coincided with its acquisition of numerous genes (green) involved in infection. Collectively, these acquired systems resulted in increased survival of *S. aureus* during interaction with host immune mechanisms, creating pathogenic *S. aureus*.

and cambialistic SODs, but Thr in iron-dependent SODs[12,29,50]. Introduction of Thr in this position in manganese-specific or cambialistic SODs increased the cambialism ratio[12,29]. This was previously explained by modification of a hydrogen-bonding network involving the mechanistically important metal-coordinated solvent[29]. Our data confirm that a Gly residue in this position strongly favors manganese-specificity, but show that a small aliphatic sidechain at this position confers cambialism when combined with an aromatic sidechain on the neighboring residue at position 160. Importantly, our structural analyses of the staphylococcal SODs and their variants did not identify changes to the hydrogen-bonding network or substrate access channel, and [1]H electron nuclear double resonance (ENDOR) spectroscopy demonstrated the positions of the solvent protons are unchanged between MnSOD and camSOD[24], and likely also in the mutants. These observations cast doubt on the hydrogen-bonding network explanation in our system, and highlight the potential of the staphylococcal SODs as a model system with which to uncover the mechanisms by which redox tuning is controlled.

Our data suggest that camSOD evolved from a MnSOD predecessor in the last common ancestor of *S. aureus*/*S. argenteus*/*S. schweitzeri*. The acquisition and subsequent altered metal utilization of camSOD coincided with the acquisition of numerous other factors that contribute to infection[8,39,41,42,51]. We exploited our ability to produce manganese-specific camSOD variants to determine whether its iron-dependent catalytic activity provides a direct benefit to *S. aureus* during exposure to stresses encountered within the host. A *S. aureus* strain expressing the Leu159Gly variant camSOD showed reduced growth under manganese-restricted and oxidative stress-exposed conditions relative to wild type. As this variant possesses reduced iron-dependent activity but increased catalysis with manganese, this result clearly demonstrates that *S. aureus* growth under these conditions requires iron-dependent camSOD activity. We previously showed that camSOD is more important than MnSOD to the establishment of infection of wild-type mice[8]. Collectively, these data support a model in which the manganese-specific camSOD antecedent, acquired by the last common ancestor of the extant *S. aureus* clade[41], underwent neofunctionalization through selection for increased iron-dependent activity, a pressure likely imposed by the manganese starvation it experiences during infection.

This study presents experimental evidence of how a change in metal specificity has naturally evolved in a metalloenzyme. We provide evidence that the specificity switch was selected for by

nutrient-starvation imposed by the immune system, implying that metal availability can shape enzyme metal utilization[18,52]. Our data also demonstrate a critical role for secondary coordination sphere residues in mediating this specificity switch, with wider implications for the study of natural and synthetic metalloenzymes.

## Methods

**Bacterial strains, culture conditions, and plasmids.** *Escherichia coli* strains DH5α and BL21(DE3) were used for cloning and for recombinant protein expression, respectively, cultured at 37 °C in lysogeny broth (LB) or M9 medium supplemented with 200 μM FeSO4, with 1.5% (w/v) agar for solid medium. *S. aureus* strains were cultured at 37 °C in tryptic soy broth (TSB), on tryptic soy agar plates (TSA), or in Tris minimal medium[53]. Where needed, tetracycline (10 μg ml⁻¹), or erythromycin (10 μg ml⁻¹), kanamycin (50 μg ml⁻¹), ampicillin (100 μg ml⁻¹), or chloramphenicol (10 μg ml⁻¹) were added to the media. All strains were stored at –80 °C in media mixed with 25% (v/v) glycerol. The construction of the *S. aureus* SH1000 Δ*sodA* (*sodA*::Tn917-LTV1), Δ*sodM* (*sodM*::*tet*), Δ*sodA*Δ*sodM* and Newman Δ*sodA* (*sodA*::*tet*) and Δ*sodM* (*sodM*::*erm*) mutant strains have been previously described[8,10]. For pull-down analysis, the C-terminally StrepII-tagged variant of camSOD (including its Shine-Dalgarno sequence) was synthesized (IDT) and sub-cloned into the pS10t shuttle vector[54] using *Bam*HI/*Pst*I restriction cloning, and the resulting pS10-Strep-sodM construct was introduced into *S. aureus* SH1000 using methods described previously[53]. Firstly, electrocompetent *S. aureus* RN4220 cells were transformed with pS10t-StrepII-sodM. Positive transformants, selected on TSB-agar plates supplemented with chloramphenicol, were used to generate φ11 phage lysate. Finally, phage transduction was used to introduce pS10t-StrepII-sodM into the *S. aureus* SH1000 background. The chromosomally integrated L159G mutation was introduced in *S. aureus* Newman via homologous recombination using the integration vector[55] pKOR1 and standard molecular approaches. This mutant was confirmed to be hemolytic by plating on blood agar prior to use.

**Expression and purification of recombinant SODs.** The cloning of the *S. aureus* *sodA* and *sodM*, their expression in *E. coli* BL21(DE3) cells, the purification of the recombinant enzymes, the in vitro swapping of bound iron for manganese by unfolding/refolding, and their quantitation, were all as previously described[8]. Expression constructs of *S. aureus* *sodA* and *sodM* were prepared by *Nde*I/*Bam*HI restriction cloning into pET29a vector. Protein was expressed in *E. coli* BL21 cells, in M9 minimal medium supplemented with 200 μM ammonium iron sulfate or in standard LB broth, for 4 h post induction with 1 mM IPTG. Cell extracts were prepared by sonication of washed cell pellets in 20 mM Tris pH 7.5, 150 mM NaCl, 1x cOmplete EDTA-free protease inhibitor (Roche), followed by centrifugation at 19,000 × g. Cleared cell lysate was subjected to chromatographic separation using an AKTA purification system (GE Healthcare). Recombinant SOD proteins were purified using anion exchange chromatography (Hi Trap Q HP column, GE Healthcare) in 20 mM Tris pH 7.5 buffer with 0–1 M NaCl gradient elution, and subsequently by size exclusion in 20 mM Tris pH 7.5, 150 mM NaCl buffer on a Superdex 200 (GE Healthcare) column. Metal cofactor exchange was achieved by unfolding SOD proteins in buffer containing 2.5 M guanidine hydrochloride in the presence of 5 mM EDTA and 20 mM 8-hydroxyquinoline to remove bound metal ions, followed by refolding through dialysis against 20 mM Tris, 100 mM NaCl, and 1–10 mM metal of interest. The constructs for recombinant expression of variants of MnSOD and camSOD were mutagenized using the QuikChange site-directed mutagenesis approach (Stratagene), using the pET29a-sodA and pET29a-sodM vectors[8] and the primer pairs listed in Supplementary Table 7.

**CD spectroscopy.** Circular dichroism data were collected using a JASCO J-810 spectropolarimeter equipped with a PTC-4235 Peltier temperature controller. Spectral measurements were performed within the far-UV range (185–260 nm) at a protein concentration of 0.8 μM in 10 mM sodium phosphate buffer, pH 7.0 at 20 °C, in a 1 mm path length quartz cuvette (Hellma), using a scan rate of 50 nm min⁻¹, 2 s response time, 0.2 nm data pitch and a 2 nm bandwidth. For each sample, five accumulated scans were averaged, corrected for the buffer reference, and then converted into mean residue ellipticity (MRE, θ). Deconvolution was performed using the CDSSTR program with reference dataset 4 (250–190 nm) on the DichroWeb server[56,57]. The thermal denaturation of protein was performed by following the change in ellipticity at a fixed wavelength of 222 nm as a function of temperature increasing from 20 to 90 °C. Protein melting temperature ($T_m$) values were calculated from the first derivative of the thermal denaturation curves, converted to mean residue ellipticity (MRE222, θ222).

**UV/visible spectroscopy.** UV/visible spectroscopy was performed on a Perkin-Elmer λ35 spectrophotometer, collecting data at room temperature between 200 and 800 nm using 10 mm path length quartz cuvettes. Measurements of resting spectra were performed in 100 mM potassium phosphate buffer, pH 7.5, 100 mM KCl, 1 mM EDTA, with samples prepared at ~800 μM protein. Spectra were converted to extinction coefficient, ε, using the protein concentration corrected for

manganese-loading, as detected by inductively coupled plasma mass spectrometry (ICP-MS). For reductive titrations, each sample was fully oxidized by incubation with 1 mole equivalent of potassium permanganate for 10 min, followed by extensive buffer exchange using centrifugal filtration devices (Amicon, 10 kDa cutoff) to remove the oxidant in the same buffer. Samples were titrated by addition of small volumes of fresh solutions of concentrated sodium dithionite, mixed vigorously by pipette and incubated for 20 min prior to acquisition of each spectrum. Spectra were corrected using the manganese content as detected by ICP-MS, and normalized to allow comparison between titrations.

**Electron paramagnetic resonance (EPR) spectroscopy.** The 94 GHz EPR spectra were obtained at 5-6 K using a Bruker Elexsys II 680 EPR spectrometer equipped with a "power upgrade 2" and an Oxford Instruments CF935 flow cryostat. The spectra were obtained by measuring the integrated amplitude of a standard two-pulse Hahn echo ($t_p(\pi) = 24$ ns and interpulse time of 400 ns) as a function of the magnetic-field. The zero-field $D$ and $E$ values were determined using a modified version of the method described in our previous work[24], because the shapes of the low-field edges of the 94 GHz spectra were different (Supplementary Note 1). To remove any arbitrariness in locating the magnetic-field of the low-field edge, the spectra were numerically differentiated and the lowest-field inflection points were taken as the edge positions. This had the effect of taking into account the distribution in the $D$ values, which produced the different edge shapes, but also overestimated their values by 100 to 200 MHz in comparison to the previously reported values. The same approach was used for the high-field edges ($E$) as well.

**ICP-MS elemental analysis.** Metal analysis was mostly performed by ICP-MS using a Thermo x-series instrument and Thermo's PlasmaLab software in Newcastle, operating in collision cell mode as previously described[8]. For Supplementary Fig. 9, ICP-MS analysis was performed by Durham University Bio-ICP-MS Facility. Briefly, protein samples were diluted in 2.5 % (w/v) solution of HNO₃ (Merck) containing 20 μg L⁻¹ Co and Pt or In and Pt (Newcastle), or 10 μg L⁻¹ Ga and Bi (Durham) as internal standards. Matrix-matched elemental standards (containing analyte metal concentrations of 0–500 μg L⁻¹) were analyzed alongside each set of samples, each in triplicate, to enable metal concentrations to be determined from standard curves.

**X-ray crystallography.** Preparations of the iron-loaded forms of recombinant MnSOD and camSOD, and the manganese-loaded forms of both double mutant variants and of the triple mutant of camSOD (15 mg ml⁻¹), were each subjected to crystallization screening and optimization as previously described[24]. Each protein preparation was confirmed to contain exclusively the target metal ion by ICP-MS prior to crystallization. Screening trays were set up using a Mosquito liquid handling robot (TTP Labtech) with commercially available matrix screens: PACT, JCSG+, Structure (Molecular Dimensions) and Index (Hampton Research) in 96-well MRC crystallization plates (Molecular Dimensions) using the sitting drop of vapor-diffusion method, incubated at 20 °C. Optimization of initial conditions for salt, polyethylene glycol (PEG) precipitant concentration and pH was performed using the hanging drop of vapor-diffusion method, incubated at 20 °C, in 24-well Linbro plates (Molecular Dimensions). Iron-loaded MnSOD was crystallized in 200 mM MgCl₂, 100 mM Tris, pH 8.5 and 30% (w/v) PEG 4000, whereas iron-loaded camSOD was crystallized in 100 mM PCTP, pH 7.0 and 25% (w/v) PEG 1500. The manganese-loaded forms of the double mutant variant of MnSOD (Gly159Leu-Leu160Phe) was crystallized in 200 mM MgCl₂, 100 mM Bis-Tris, pH 5.5 and 25% (w/v) PEG 3350, whereas the manganese-loaded form of the double mutant variant of camSOD (Leu159Gly-Phe160Leu) was crystallized in 30 mM MgCl₂, 30 mM CaCl₂, 39.1 mM Bicine, 60.9 mM Trizma, pH 8.5, 20% (w/v) PEG 550 MME and 10% (w/v) PEG 20,000. The manganese-loaded form of the triple mutant variant of camSOD (Ile19Phe-Leu159Gly-Phe160Leu) was crystallized in 100 mM potassium thiocyanate and 30% (w/v) PEG 2000 MME. All but the double mutant camSOD were cryo-protected with 20% (w/v) PEG 400. X-ray diffraction data collection (Diamond Light Source, Didcot, UK) and processing, and model building and validation were all as previously described[24], except that data processing also used xia2 and phasing also used Phaser[58]. The search model for iron-loaded wild-type MnSOD and manganese-loaded MnSOD Gly159Leu-Leu160Phe was PDB model 5N56. The search model for iron-loaded wild-type camSOD and the manganese-loaded forms of camSOD Leu159Gly-Phe160Leu and SodM Ile19Phe-Leu159Gly-Phe160Leu was PDB model 5N57. The PDB codes for the structures of the iron-loaded forms of MnSOD and camSOD and the manganese-loaded forms of the camSOD triple mutant variants are 6EX3, 6EX4, and 6EX5, respectively. The data collection and refinement statistics are summarized in Supplementary Table 3. All crystallographic images were generated using PyMOL Molecular Graphics System, Version 1.8 (Schrödinger, LLC). To illustrate structural regions where the sequence is conserved, ConSurf was used[59] to analyze 500 HMMER homologs of the SOD family, sampled from the UniREF90 database. Conservation scores were calculated using the Bayesian method.

**Pull-down assay.** The StrepII-tagged camSOD was constitutively expressed in a *S. aureus* SH1000 Δ*sodM* strain[10] and purified using affinity chromatography. Cell

extracts were prepared by freeze-grinding under liquid nitrogen in 20 mM Tris, pH 8.0, 150 mM NaCl, 1 mM EDTA. Thawed, ground material was centrifuged ($4000 \times g$, 20 min, 4 °C) to remove cell debris, and the resulting supernatant was filtered through a 0.45 μm membrane filter. The soluble protein extract was applied to a StrepTrap column (GE Healthcare), equilibrated and subsequently washed in 20 mM Tris pH 8.0, 150 mM NaCl, 1 mM EDTA. Elution (1 ml) used 2.5 mM desthiobiotin (PanReac AppliChem) in this buffer. Fractions were analyzed by SDS-PAGE and ICP-MS.

**SOD activity assays.** SOD activity was assessed qualitatively in-gel, or quantitatively using a SOD Assay Kit (Sigma-Aldrich) per the manufacturer's instructions, as previously described[8]. S. aureus cell extracts were prepared by mechanical disruption in 50 mM phosphate buffer at pH 7.8 with 0.1 mM EDTA. Protein content of lysates was quantified by bicinchoninic acid assay (Sigma). Aliquots of crude extracts (5.4 μg) or purified recombinant proteins (0.3 μg) were resolved on 10% (w/v) native polyacrylamide gels. For staining, gels were incubated in 50 mM potassium phosphate buffer pH 7.8, 1 mM EDTA, 0.25 mM nitroblue tetrazolium chloride (Sigma), 50 μM riboflavin and then exposed to light to detect SOD activity. Incubation of replica gels in 20 mM $H_2O_2$ prior to activity staining was performed to assess Fe-dependent SOD activity. Note that, as the native gels used for assays were stained with a SOD-specific stain, no molecular weight markers were resolved on these gels.

**Calprotectin growth assays.** Assays of S. aureus growth in the presence of calprotectin and paraquat were performed as previously described[8]. Overnight S. aureus cultures were set up in Chelex-treated RPMI medium containing 1% casamino acids (NRPMI), supplemented with 1 mM $MgCl_2$, 100 μM $CaCl_2$, and 1 μM $FeCl_2$. These were further sub-cultured by 1:100 dilution into 100 μl of fresh culture medium consisting of 38% (v/v) TSB and 62% (v/v) CP buffer (3 mM $CaCl_2$, 20 mM Tris base, 100 mM NaCl, 10 mM β-mercaptoethanol, pH 7.5) supplemented with 1 μM $MnCl_2$ and 1 μM $ZnSO_4$. Growth assays were performed in a 96-well round-bottom plate at 37 °C with shaking at 180 rpm for 8 h. Where indicated, 0.1 mM PQ was added to the media.

**SOD phylogenetic analysis.** SOD sequences were identified using BLASTP search[60] (E-value threshold = 0.005), with S. aureus MnSOD sequence as query, against a database of protein sequences downloaded from the NCBI server encoded in 2577 bacterial (including Candidate Phyla Radiation[61,62]), 289 archaeal (including ASGARD Archaea[63,64]), and 148 eukaryotic genomes. All 2691 identified SOD homologs were aligned using MAFFT[65] and trimmed with trimAl[66] (gappyout mode) resulting in a trimmed alignment of 208 positions (Supplementary Data 1). The best fitting phylogenetic model (WAG + G) was selected with PROTTEST[67], and used to generate a SOD phylogeny in RAXML[68] with 100 bootstrap replicates. The tree was visualized and annotated with the coevolving amino acid residues (Fig. 4 and Supplementary Fig. 9) using GraPhlAn[69] (Supplementary Data 1 and 2).

**Amino acid correlation analysis.** Groups of coevolving amino acids were identified (Supplementary Data 2) in the alignment of the 2,691 SOD sequences sampled across the tree of life (Supplementary Data 1), using amino acid correlation algorithm implemented in PFstats[70] following the user guide. PFstat correlation analyses parameters were as follows: Minimum Score: 60 (18.75% of the maximum positive correlation score of 320); MRsA fraction: 0.2; minimum delta: 0.3. The amino acid correlation network was visualized in Cytoscape[71] (Supplementary Fig. 9).

**Protein network analyses.** Protein similarity networks (Supplementary Data 3) were generated out of 71,757 protein sequences encoded in 29 analyzed Staphylococcal genomes (downloaded from the NCBI server) with BLAST[60], and quick edge file optimization implemented in EGN[72] using the following thresholds: BLAST ($a = 12$, $e = 1e^{-05}$), simple link parameter (E-value threshold = $1e^{-05}$, hit identity threshold = 20%, minimum hit coverage = 20%, without the best reciprocal hit condition enforcement), and with the quick edge file creation (E-value threshold = $1e^{-05}$, hit identity threshold = 20%, minimum hit coverage = 20%, minimal hit length = 30 amino acids). Protein sequences were annotated[73] using eggNOG-mapper v1. The generated networks were analyzed using igraph[74] and dnet[75] packages in R[76], and visualized in Cytoscape[77]. Analyzed species were classified into two categories: camSOD-positive (S. argenteus MSHR1132, S. aureus COL, S. aureus MU50, S. aureus Newman, S. aureus USA300LV CA12, S. aureus USA300 FPR3757, S. aureus NCTC8325); camSOD-negative (S. capitis AYP1020, S. carnosus SK13, S. chromogenes MU970, S. cohnii SNUDS2, S. epidermidis ATCC12228, S. equorum C2014, S. haemolyticus JCSC1435, S. hominis K1, S. hyicus, S. intermedius NCTC11048, S. lentus 050AP, S. massiliensis CCUG55927, S. microti, S. pseudintermedius NA45, S. saprophyticus ATCC15305, S. schleiferi 2317 03, S. sciuri FDA ARGOS285, S. simiae NCTC3000, S. simulans FDA ARGOS383, S. vitulinus F1028, S. warneri SG1, S. xylosus HKUOPL8). The Identified in all networks were defined as those networks containing only a single homolog in each of the analyzed genomes; not identified in S. aureus networks were defined as

networks not containing protein homologs from any of the camSOD-positive genomes, and containing homologs from at least 11 out of 22 camSOD-negative species; S. aureus-specific networks were defined as networks containing protein homologs from 7 out of 7 genomes from camSOD-positive species, and none from the camSOD-negative species. To be defined in the group labeled as Enriched in S. aureus, each camSOD-positive species had to contain more protein homologs than any of the camSOD-negative species within the network. Heatmaps of the identified protein networks (Fig. 5 and Supplementary Fig. 12) were generated using gplots package (https://CRAN.R-project.org/package=gplots) within R.

**Staphylococcus species tree.** To generate a species tree of the analyzed Staphylococci (Supplementary Data 1), single orthologues from 104 Identified in all networks (Supplementary Data 4), were aligned using MAFFT[65], trimmed with trimAl[66], and concatenated with module Nexus of the Biopython[78] package to a final alignment of 24,091 amino acid sites. The best fitting phylogenetic model was selected using ModelFinder[79] implemented on the CIPRES Science Gateway V. 3.3 web server[80], according to the BIC criteria. Phylogenetic tree was generated under LG+F+R3 model with 1000 ultrafast bootstrap replicates using IQ-TREE[81] implemented on the CIPRES Science Gateway V. 3.3 web server[80].

**Circular genome comparison.** Circular genome comparison was generated with BRIG[82], using S. aureus NCTC8325 genome[83] as a reference, which was compared to 29 analyzed Staphylococcus genomes using BLASTN[60] (Fig. 5). S. aureus NCTC8325 virulence factors (PATRIC[84], and VFDB[85] databases), and essential genes (PATRIC[84] and AureoWiki[38] databases) annotations (Supplementary Data 5) were downloaded from the respective databases on 29th March, 2019.

**Protein network analyses of metal acquisition systems.** Networks containing homologs of proteins known to be involved in manganese and iron import were isolated using igraph[74] and dnet[75] packages in R[76], and visualized in Cytoscape[77] (Supplementary Fig. 12). Sequence of the proteins present in the networks were aligned using MAFFT[65], and the alignments were trimmed with trimAl[66] and used to generate phylogenies under LG + G model in FastTree[86]. Homologs of the proteins of interest were identified based on the network and tree topologies, together with manual inspection of protein sequence alignments, and used to generate the heatmap with gplots package.

**Other bioinformatic methods.** Multiple sequence alignments were visualized and annotated using Jalview[87]. Phylogenetic trees were visualized and annotated using FigTree (http://tree.bio.ed.ac.uk/software/figtree/), Archaeopteryx[88], and BRIG[82]. Linear genome fragments were aligned and visualized using EasyFig[89].

**Reporting summary.** Further information on research design is available in the Nature Research Reporting Summary linked to this article.

## Data availability
Structural data that support the findings of this study have been deposited in the Protein Data Bank with the accession codes PDB 6EX3, PDB 6EX4, PDB 6EX5, PDB 6QV8, and PDB 6QV9 (see Supplementary Table 2). Source data for Table 1, Fig. 6 and Supplementary Fig. 1b–d, f are provided with the paper. There are no restrictions on any data within the manuscript. Biological materials will be provided on request.

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

## Acknowledgements

K.J.W. was supported by a Wellcome Trust- and Royal Society-funded Sir Henry Dale fellowship (098375/Z/12/Z). A.B.-S. was supported by a BBSRC Ph.D. studentship and funding from Newcastle University's Faculty of Medical Sciences, except during a revision period when she was funded by a Wellcome Trust Collaborative Award (204877/Z/16/Z). E.S.M. was also supported by a BBSRC Ph.D. studentship. A.B. was funded by Newcastle University's Faculty of Medical Sciences. T.E.K.-F. was supported by operating grants from the National Institutes of Health (R01 AI118880) and the Vallee Foundation. The high-field EPR measurements benefited from the biophysics platform of the Institute for Integrative Biology of the Cell, supported by the French Infrastructure for Integrated Structural Biology (FRISBI) ANR-10-INBS-05. We thank Diamond Light Source for access to beamline I03, I04, and I24 (mx7864, mx9948, and mx18598). The contents of this work are solely the responsibilities of the authors and do not reflect the official views of any of the funders, who had no role in study design, data collection, analysis, decision to publish, or preparation of the manuscript.

## Author contributions

A.B.-S. designed and performed most in vitro experiments, with further contributions from E.S.M., E.T., and P.C. Y.M.G. performed *S. aureus* experiments. A.B. assisted A.B.-S. with X-ray crystallography, and K.M.S. performed all bioinformatic analyses with assistance from C.B. for network analysis. S.U. performed HFEPR experiments and analyzed these data, with input from L.C.T., who also prepared the auto-oxidation calculations. K.J.W. and T.E.K.F. conceived and managed the study, and wrote the manuscript with S.U., with input from all authors.

## Competing interests

The authors declare no competing interests.
