## [Peer Review File · Nature Communications]

Reviewers' comments:

Reviewer #1 (Remarks to the Author):

In this paper, the authors present a study on *S. aureus* Fe/Mn- and Mn-dependent SODs, showing that the different metal specificity of the two enzymes depends on subtle structural factors located to the second coordination sphere of the metal ion. The work is well designed and the results are a valuable contribution to research in the field of bioinorganic chemistry and beyond. I thus expect that this paper will obtain broad interest throughout the scientific community. Nevertheless, I think that the authors somehow overstated the novelty of their results: specifically, at the end of discussion they state that their study “represents the first experimental evidence of how a change in metal specificity has naturally evolved in a metalloenzyme”, and that they demonstrate “a critical and surprising role for secondary coordination sphere residues in mediating this specificity switch”. In fact, the evolution of metal specificity in metalloenzymes has already been addressed several times in the past; also, the importance of the second coordination sphere in modulating the physico-chemical properties of metal-binding sites in proteins, including metal identity, is a steady concept in bioinorganic chemistry. Some examples can be found in, e.g., *J Biol Inorg Chem.* 2019 Jun;24(4):517-531, *J Inorg Biochem.* 2018 Feb;179:40-53, *J Biol Chem.* 2015 Jul 31;290(31):18914-23, *Acc Chem Res.* 2014 Oct 21;47(10):3110-7, *Protein Sci.* 2010 Jan;19(1):75-84, *Biochemistry.* 2005 Jun 14;44(23):8378-86

Reviewer #2 (Remarks to the Author):

I have read the manuscript entitled An evolutionary path to altered cofactor specificity in a metalloenzyme by Barwinska-Sendra et al and have a number of comments.

The ability to examine the evolution of the two SODs and say something about metal use and metal stress is remarkable. To introduce mutations into an enzyme and alter the specificity of metal utilization is also impressive. With regards to the latter point, however, I do have some comments.

There was a paper published in 2018 (Hunter et al, *Chem. Eur. J* 2018 24 5303) that reported on a single mutation that changed the metal specificity of the MnSOD in *C. elegans* such that it became cambialistic. The modification was not to a metal-binding ligand but was in the second sphere. It is not analogous to modification reported here as there were proton interactions but it does warrant mentioning as a single mutation has changed metal specificity.

The FeSOD has been shown in most cases to carry out a robust Fenton-type reaction with the peroxide formed in the dismutase reaction. This either leads to enzyme deactivation or to formation of a powerful oxidant in solution. The MnSOD has not been shown to carry out any of this

peroxidative chemistry. This has certainly led to some speculation as to why certain species use certain of these SOD in general and is also worth noting here even though it is more obscure in the *S. aureus*.

In Table 1, in all cases the enzymes with high cambialistic ratios have overall lower activity than the enzymes that are specific for a metal. This is a truism for virtually all naturally occurring cambialistic enzymes; there is a loss in the overall activity associated with the ability to bind either metal and have activity. I don't believe that this is necessarily associated with changes in auto-oxidation of cambialistic Mn/FeSODs. The autooxidation results and speculated connection with redox potential changes is, for me, particularly significant and worth discussing in a little more detail.

Overall, I find this to be a compelling paper.

Reviewer #3 (Remarks to the Author):

In this study, Anna Barwinska-Sendra and co-workers report on the characterization of point-mutants in superoxide dismutases (SODs) that alter their relative metal reactivity. The authors chose two SODs from *S. aureus*, the former corresponding to a Mn-specific SOD and the latter to a cambialistic one (i.e. able to react either using Fe or Mn as metal cofactor). They combined amino acid sequence and X-ray structure comparisons to identify three sites likely responsible for explaining such changes in metal reactivity. They subsequently swapped amino acids from one to another SOD and observed indeed changes in metal reactivity that they attributed to changes in redox potentials due to changes in the second sphere of coordination environment. Determining the factors that can alter metal cofactor reactivity are important for both medicine and biotechnological applications. Overall the work is robust and well conducted. The authors have carefully checked, using X-ray crystallography at high-resolution, that the mutations do not affect either the whole structure or the metal environment. They provide an extensive bioinformatic analysis to complete their work showing that most likely, the cambialistic SODs derive from Mn-specific ones, which would have further evolved after a gene duplication event. However, the study lacks metal binding affinity constant measurements to better characterize the effect of these mutations. A more systematic approach to identify which amino acid would be the best ones at these positions to either select Mn- or Fe-dependent activity. The "cambialistic" ratio is indeed informative, but for some of the mutants, the drop in the overall activity seems more important than the change in metal-reactivity. Indeed, while the Fe-dependent activity increases (from a low level) the Mn-dependent one seems to collapse, leading to a better "cambialistic" ratio that may be artificially overestimated. The authors have clearly identified a very interesting factor but the reviewer remains a bit disappointed because one would have expected some explanations about why/how changing these residues alter the metal reactivity. For example new redox potential determinations would have been appreciated. What is really modified when changing the second coordination sphere? This information would have been valuable for future applications in developing synthetic

metalloenzymes. In the reviewer's opinion, the manuscript does not fit the broader readership of Nature Communication and would be more suitable to a specialized journal.

Minor comments:

The authors refer to Mn(II) oxidation state in line 197, but it sounds more to electronic structure or redox potential because as it is Mn(II) the oxidation state cannot be modified unless it is not Mn(II) any more. Please clarify.

The authors often refer to metal specificity. The reviewer would suggest metal reactivity because metal specificity would better correspond to either Mn or Fe preferentially bound to the active site. Please clarify this point.

Reviewer #4 (Remarks to the Author):

This paper aims to identify key structural and physical properties changes that allow SOD enzymes and mutants thereof from a bacterium to utilize two different ions, Mn and Fe. HF EPR is used to determine the zero-field interaction of the various mutants which is very sensitive to the electronic structure and coordination symmetry. The spectral differences are shown in Figure 3a which are small but certainly significant. The authors then cite a paper where in model compounds these differences in zero-field interaction (D and E) are correlated with redox potential. As this is an enzyme and the paper aims to characterise structural and physical differences, the actual redox potentials should be measured to confirm this is actually true in this case. This data could be obtained by an EPR redox titration, clearly the Authors demonstrate that the EPR signal is quite strong and so this type of EPR experiment should be feasible (at X-band or W-band) even if a bit tedious. The Authors state in the conclusions and elsewhere that the altered redox potential is a main cause of the change in reactivity, but it was never measured. The uncertainty in this model complex correlation to this enzyme is highlighted by Supplementary Figure 5 where D/E ratios do not fit the model compounds. The redox potentials need to be measured for the paper to be impactful, as the Authors write in their conclusion on page 13, line 319: 'enabling future studies to determine how the architecture of a metalloenzyme imposes cofactor specificity'. Meaningful redox potentials would be very important for this future aim.

Minor points

On page 7, line 63, the Authors state that the positions of the mutated residues are not altered. This is at first a confusing statement because they must mean the position of the backbone as the side-

chains are clearly different. This sentence and the next ones (like page 7, line 176 & page 8, line 194) should be rewritten. In this regard, the accuracy of the X-ray structure should be discussed, in particular the bond lengths and angles around the Mn ion (i.e. describe briefly Supplementary Table 4 and the errors).

Page 8, line 186. 'zero field interaction', zero-field is one word and needs the hyphen

Figure 3A and tables Supplementary 5: The HFEPR spectra were analyzed using second-order expressions, which should be accurate enough to extract the D and E values (Supplementary Table 5). However, it would be beneficial to the reader to include a simulation of each spectrum using a full matrix diagonalization approach. This is a very easy task today and numerous free software packages are available to achieve this. These spectra should appear in the Supporting Material.

We wish to thank all four reviewers for their time. It is clear from their comments that they have thought carefully about the data we presented in this manuscript, and they raise a number of important points about our study for consideration. We address each of these point-by-point below:

Reviewer #1:

*In this paper, the authors present a study on *S. aureus* Fe/Mn- and Mn-dependent SODs, showing that the different metal specificity of the two enzymes depends on subtle structural factors located to the second coordination sphere of the metal ion. The work is well designed and the results are a valuable contribution to research in the field of bioinorganic chemistry and beyond. I thus expect that this paper will obtain broad interest throughout the scientific community.*

1. We thank the reviewer for their positive comments and we agree that this manuscript should obtain broad interest once published.

Nevertheless, I think that the authors somehow overstated the novelty of their results: specifically, at the end of discussion they state that their study “represents the first experimental evidence of how a change in metal specificity has naturally evolved in a metalloenzyme”, and that they demonstrate “a critical and surprising role for secondary coordination sphere residues in mediating this specificity switch”. In fact, the evolution of metal specificity in metalloenzymes has already been addressed several times in the past; also, the importance of the second coordination sphere in modulating the physico-chemical properties of metal-binding sites in proteins, including metal identity, is a steady concept in bioinorganic chemistry. Some examples can be found in, e.g., J Biol Inorg Chem. 2019 Jun;24(4):517-531, J Inorg Biochem. 2018 Feb;179:40-53, J Biol Chem. 2015 Jul 31;290(31):18914-23, Acc Chem Res. 2014 Oct 21;47(10):3110-7, Protein Sci. 2010 Jan;19(1):75-84, Biochemistry. 2005 Jun 14;44(23):8378-86.

2. We agree with the reviewer that the fact that such changes in metal specificity do occur over evolutionary time has been reported previously. Two reviews cited by the referee, as well as many other publications (Kirschvink *et al.*, 2000, Andreini *et al.*, 2008, Dupont *et al.*, 2010, Uberto & Moomaw, 2013), have described the observation of such evolutionary changes in metal utilization. However, these prior studies have relied on phylogenetic analysis to identify such evolutionary changes. The novelty of our study was encapsulated within the sentence referred to by the reviewer. Specifically, this work demonstrates experimentally how specific mutations to residues within the metal's secondary coordination sphere that have been selected throughout the evolution of a metalloenzyme have resulted in altered metal specificity. We do appreciate the reviewer's concern though, and have modified the paragraph of the Introduction in which we described these evolutionary changes in metal usage to accommodate this point, adding a key reference cited by the reviewer (Valasatava *et al.*, 2018) and pointing out there was prior bioinformatic support for this concept (lines 72-3, page 4).

We also agree with the reviewer that modulation of the physicochemical properties of a bound metal by amino acid residues within the secondary coordination sphere is a well established concept. Indeed, previous biochemical and biophysical studies of the SOD enzymes have played an important role in the development of the concept termed 'redox tuning' by metalloenzymes (see multiple further comments on this point, below) (Hsieh *et al.*, 1998, Vance & Miller, 1998, Vance & Miller, 2001, Sjodin *et al.*, 2008, Miller, 2008, Grove *et al.*, 2008, Sheng *et al.*, 2014). Nonetheless, this remains a unique study in that it has demonstrated that such modulation of secondary coordination sphere residues has been selected for during evolution of the *S. aureus* SODs to convert MnSOD into camSOD. Specifically, by making reciprocal mutations in our closely-related pair of staphylococcal SOD metalloenzymes that have divergent metal specificities, we converted an ancestral manganese-dependent enzyme (MnSOD) into a cambialistic enzyme (camSOD) that is able to utilize either manganese (Mn) or iron (Fe). Furthermore, by making reciprocal mutations, we demonstrated that the sequence changes that drove the emergence of the camSOD provide an advantage to *Staphylococcus aureus* in resisting naturally occurring stressors encountered during

infection (manganese starvation and oxidative stress). In conjunction with our previous publication (Garcia *et al.*, 2017), we have demonstrated that this change in metal specificity is physiologically relevant and plays an important role in enabling this pathogen to maintain its oxidative stress defence during infection, when it experiences manganese starvation. Thus, our study has demonstrated how evolution was able to switch metal specificity by altering secondary coordination sphere residues. This is in contrast to the cited examples, where changes in metal specificity (or indeed metal selectivity, a different concept related to the propensity of a metalloenzyme to selectively bind rather than to specifically catalyse its metal-dependent reaction – see further comments on this in point 15, below) were induced *in vitro* by introducing mutations that were not identified in nature into existing metalloenzymes to artificially alter their specificity. To avoid overstating our work, we have removed ‘surprising’ from the sentence noted by the reviewer on lines 416-8, page 16. To more clearly and explicitly state the importance of our study, we have also re-phrased the final sentence of the first paragraph of the Discussion, which now reads “Exploiting this platform demonstrated that evolution can effect substantial changes in metal specificity by making modest changes in the chemical nature of secondary coordination sphere residues that regulate metal reactivity” (lines 359-61, page 14, Discussion).

Reviewer #2 (Remarks to the Author):

I have read the manuscript entitled An evolutionary path to altered cofactor specificity in a metalloenzyme by Barwinska-Sendra et al and have a number of comments.

The ability to examine the evolution of the two SODs and say something about metal use and metal stress is remarkable. To introduce mutations into an enzyme and alter the specificity of metal utilization is also impressive. With regards to the latter point, however, I do have some comments.

There was a paper published in 2018 (Hunter et al, Chem. Eur. J 2018 24 5303) that reported on a single mutation that changed the metal specificity of the MnSOD in C. elegans such that it became cambialistic. The modification was not to a metal-binding ligand but was in the second sphere. It is not analogous to modification reported here as there were proton interactions but it does warrant mentioning as a single mutation has changed metal specificity.

3. The publication to which the reviewer refers involves a mutation introduced into the MnSOD-3 isozyme from the worm *Caenorhabditis elegans*, which resulted in altered metal specificity. However, the crucial difference between their study and ours was that the mutation they introduced was based on a comparison of the sequence of this enzyme from this multicellular eukaryote with that of the FeSOD from the bacterium *Mycobacterium tuberculosis*. These enzymes are distantly related (<43% sequence identity) and from different branches of the evolutionary tree, whereas in our study we reciprocally switched residues between very closely related isozymes (75% identity) from the same species. Specifically, our system enables us to conclude that evolution has switched the metal specificity of the staphylococcal camSOD by selecting mutations of these residues within the secondary coordination sphere, rather than simply that mutations in this sphere can result in changes in specificity, as shown in previous publications. We agree that the paper they refer to is worthy of citing in our manuscript. We have added it as further support for our statement ‘These results support a model in which the SOD architecture selectively tunes the cofactor’s reduction potential’ on lines 374-5, page 15 of the Discussion, and have added this reference to the bibliography (reference no. 44).

We agree with the reviewer that the mutation identified by Hunter *et al.* is not analogous to the mutations identified in our current work. The mutation used in the Hunter study was Q142H, which, as the reviewer explains, results in changes to the hydrogen-bonding network in the active site. Notably, our naturally-occurring mutations do not appear to have significantly interrupted the hydrogen-bonding network based on the crystal structures of our variants, which is why we proposed the reduction potential changes as the likely mechanism, consistent with extensive SOD literature and the redox tuning model (see further comments in points 12 and 16, below). It also

sets our study apart from previous work on SOD metal specificity that have observed changes in the hydrogen-bond network when metal specificity changes.

The FeSOD has been shown in most cases to carry out a robust Fenton-type reaction with the peroxide formed in the dismutase reaction. This either leads to enzyme deactivation or to formation of a powerful oxidant in solution. The MnSOD has not been shown to carry out any of this peroxidative chemistry. This has certainly led to some speculation as to why certain species use certain of these SOD in general and is also worth noting here even though it is more obscure in the S. aureus.

4. We agree that iron-dependent enzymes are prone to Fenton-type chemistry, while manganese-dependent enzymes typically are not. Thus, as a whole it is certainly a possibility that the threat of oxidative stress has driven enzymes toward manganese utilization. However, in the case of our system, iron utilization has been evolutionarily selected, as *S. aureus* utilizes the iron-loaded camSOD to maintain a defence against superoxide when manganese-starved by the host-during infection (Garcia *et al.*, 2017). While we agree with the reviewer, due to space constraints and the untested nature of this chemistry in our system, we have not directly commented on this issue in the manuscript.

In Table 1, in all cases the enzymes with high cambialistic ratios have overall lower activity than the enzymes that are specific for a metal. This is a truism for virtually all naturally occurring cambialistic enzymes; there is a loss in the overall activity associated with the ability to bind either metal and have activity.

5. We agree with the reviewer on this very important point. Having analysed a large volume of data from previous SOD publications in order to annotate the 'known' metal specificities of the isozymes represented in our bioinformatics analyses (see Fig. 4), we also noted that this trend holds for most cambialistic enzymes. Metal-specific SOD enzymes are clearly more efficient catalysts than cambialistic SOD enzymes, almost always exhibiting a higher turnover. We suggest that cambialistic SODs represent an evolutionary compromise, where catalytic activity is sacrificed at the expense of enabling cofactor flexibility. Indeed, this is presumably the reason why organisms such as *Escherichia coli* possess dual genes encoding cytosolic SOD isozymes – one specific for manganese and the other specific for iron. If cambialistic enzymes could be optimized by evolution to gain the level of activity achieved by the metal-specific SODs while maintaining cofactor flexibility, then these organisms would have no need to encode two separate genes. Nonetheless, it's important to note that we have previously demonstrated that camSOD (SodM) is important, despite its low enzymatic turnover, but MnSOD (SodA) is not, during *S. aureus* infection of wild type mice *in vivo* (Garcia *et al.*, 2017). This observation highlights that maximizing catalytic activity is not the only aspect of metalloenzyme function that evolution can optimize. To clarify this in the manuscript, we have added two sentences about this evolutionary compromise to the Results (lines 99-102 and 163-6 on pages 5 and 7, respectively), and a sentence about the key role of camSOD *in vivo* to the Discussion (lines 406-7, page 16).

I don't believe that this is necessarily associated with changes in auto-oxidation of cambialistic Mn/FeSODs. The autooxidation results and speculated connection with redox potential changes is, for me, particularly significant and worth discussing in a little more detail.

6. We thank the reviewer for their appreciation of the auto-oxidation results we presented. In light of this and other comments made by the reviewers, we now appreciate that we had not previously done the best job possible of integrating our biochemical, structural and biophysical data to explain our interpretation of the relationship between these auto-oxidation data and the metal's reduction potential. We have now re-written the relevant sections of the Results (lines 179-252, pages 8-10) to explain this relationship more clearly. We have added new data showing differences in the ability of the reductant dithionite to reduce the two SOD enzymes and their mutated variants (see further comments below in points 12 and 16), and we also present calculations in the Supplementary

Information, illustrating the theoretical relationship between the auto-oxidation results and the metal's reduction potential.

Overall, I find this to be a compelling paper.

7. We appreciate the reviewer's positive response to our manuscript and hope that the modifications we have made in response to the important points that all of the reviewers have raised have resulted in an improved document overall.

Reviewer #3 (Remarks to the Author):

*In this study, Anna Barwinska-Sendra and co-workers report on the characterization of point-mutants in superoxide dismutases (SODs) that alter their relative metal reactivity. The authors chose two SODs from *S. aureus*, the former corresponding to a Mn-specific SOD and the latter to a cambialistic one (i.e. able to react either using Fe or Mn as metal cofactor). They combined amino acid sequence and X-ray structure comparisons to identify three sites likely responsible for explaining such changes in metal reactivity. They subsequently swapped amino acids from one to another SOD and observed indeed changes in metal reactivity that they attributed to changes in redox potentials due to changes in the second sphere of coordination environment. Determining the factors that can alter metal cofactor reactivity are important for both medicine and biotechnological applications. Overall the work is robust and well conducted. The authors have carefully checked, using X-ray crystallography at high-resolution, that the mutations do not affect either the whole structure or the metal environment. They provide an extensive bioinformatic analysis to complete their work showing that most likely, the cambialistic SODs derive from Mn-specific ones, which would have further evolved after a gene duplication event.*

8. We would like to thank the reviewer for their appreciation of the robust, well-conducted experiments in this manuscript, the rigor involved in verifying that the structures of our enzymes are essentially unchanged in our mutant SODs, and the extensive nature of our supporting bioinformatic analyses, which we feel will also prove to be a valuable resource for the SOD research community in addition to providing key evidence for our study.

However, the study lacks metal binding affinity constant measurements to better characterize the effect of these mutations.

9. We would very much like to measure the metal binding affinities of these enzymes, and not just for completeness of our characterization of these enzymes and their mutants. We, and others, have shown that the SODs are competent to bind a number of divalent metal ions (see Supp. Fig. 1). It is therefore anticipated that these enzymes could be used as a tool to report on cytosolic metal availabilities under varying conditions, if the binding affinities for multiple metals were known. Unfortunately, however, determination of dissociation constants is not possible with these enzymes at present. The metal cofactor is deeply buried within the enzyme fold, such that the metal off-rate after folding is essentially zero. Switching of the bound metal ion *in vitro* requires us to unfold the protein in the presence of concentrated chaotropic agents, to remove the metal using a combination of high affinity metal chelators, and then to refold the protein in the presence of the desired target metal ion. This process is performed by dialysis over the course of several days. The apo- form of the protein (after refolding in the absence of metal and in the presence of high affinity metal chelators) is unstable and prone to degradation, precluding experiments with it to determine metal binding thermodynamics or kinetics. Taken together, this suggests that the metal ion likely binds to the enzyme when it is in a partially folded state, presumably associating during the enzyme's folding pathway *in vivo*. Therefore it is impossible to measure a true dissociation constant, which would require equilibrium between bound and unbound states to be achieved. We have added a sentence to the legend for Supp. Fig. 1, which is the only figure in the manuscript that deals with metal selectivity, to state this technical issue explicitly, stating "Note that the kinetic trapping of the metal ion by the enzymes, and the instability of the apo-forms of the proteins (data not shown) have thus far precluded experimental affinity measurements" (SI, lines 47-9, page 2).

While metal binding affinity measurements would be informative for the field at large, we believe that their absence does not detract from this manuscript. This is because they would provide information about metal selectivity, whereas this study elucidates how evolution has shaped metal specificity (see further comment number 15, below). We have previously demonstrated that the camSOD acquires an iron cofactor in *Staphylococcus aureus* cells under conditions of manganese starvation. Therefore the current work has focused on how the Fe-form of this enzyme functions and the evolutionary path that enabled the conversion of the MnSOD to the camSOD, which is capable of utilizing Fe as a cofactor, rather than how it acquires that cofactor.

A more systematic approach to identify which amino acid would be the best ones at these positions to either select Mn- or Fe-dependent activity.

10. As explained in response to reviewer 1 in our comment number 2, above, the novelty of our study is that this manuscript demonstrates experimentally how specific mutations that have been selected throughout the evolution of a metalloenzyme have resulted in altered metal specificity of the encoded enzyme. We achieved this by dissecting the results of evolution using bioinformatics, and by structurally and biochemically characterizing the effects of reciprocally swapping the residues near to the active site that differ between the MnSOD and the camSOD. Thus we did not set out with the goal of artificially converting a manganese-specific SOD into a cambialistic or iron-specific SOD but in interrogating the effects of the naturally-occurring mutations that evolution had introduced during divergence of camSOD and MnSOD in *S. aureus*.

This being said, we agree with the reviewer that it would be very interesting, now that we have defined sequence positions that modulate metal specificity, to next use a systematic approach to determine how the chemical properties of the amino acid residues at these positions influence the enzyme's metal specificity. However, such a focused systematic analysis would require a quantity of work comparable to that presented in the current manuscript. Therefore we believe that the suggested experiments warrant an independent study.

The “cambialistic” ratio is indeed informative, but for some of the mutants, the drop in the overall activity seems more important than the change in metal-reactivity. Indeed, while the Fe-dependent activity increases (from a low level) the Mn-dependent one seems to collapse, leading to a better “cambialistic” ratio that may be artificially overestimated.

11. We largely agree with the reviewer on this point. It seems that cambialistic enzymes represent a level of compromise between maximising catalytic rate vs. enabling cofactor flexibility and in some instances, such as with the camSOD, flexibility is the driving factor (see comment 5, above). As a general rule, mutations in a MnSOD that increase the activity with an Fe cofactor necessarily decrease its activity with a manganese cofactor, an observation that is consistent with the redox tuning model of SOD function (Miller, 2008, Sheng *et al.*, 2014). Indeed, this is one of the reasons why we felt it was important to define the ‘cambialism ratio’ (CR) in this manuscript as a new quantitative measure of the level of cambialism, to enable SOD enzymes to be more easily compared. We disagree, however, with the interpretation that this could be described as ‘artificially overestimated’, as the data (especially when presented as both absolute units and cambialism ratio) illustrate this phenomenon of compromise between specificity and activity. Nonetheless, we appreciate the reviewer's comment as it identifies the very reason why we have presented both the absolute activity and the CR in Table 1. To address this in the revised manuscript, we have added a clause to the description of the cambialism ratio (line 135, page 6), and a sentence to the Results section that describes the activity and CR of the mutated variants (lines 166-7, page 7) that specifically states the importance of comparing both the absolute activity and the cambialism ratio when assessing metal specificity.

The authors have clearly identified a very interesting factor but the reviewer remains a bit disappointed because one would have expected some explanations about why/how changing these residues alter the metal reactivity. For example new redox potential determinations would

have been appreciated. What is really modified when changing the second coordination sphere? This information would have been valuable for future applications in developing synthetic metalloenzymes.

12. We agree that reduction potential measurements would be very informative to understand the molecular mechanism by which the secondary coordination sphere residues regulate metal specificity, and have made further experimental efforts to clarify this point for the reviewers. We also recognize that we did not satisfactorily integrate our description and discussion of the multiple lines of experimental evidence (biochemical, structural and biophysical) that have led us to the conclusion that changes in reduction potential mechanistically regulate metal specificity in the original manuscript. We have edited the text to better make these points.

Despite extensive effort, we have been unable to generate sufficiently quantitative electrochemical data to estimate the midpoint reduction potentials of our SOD enzymes (see below). However, we have generated new qualitative data (Supplementary Figure 7) that support our interpretation that the enzymes' reduction potentials differ. Specifically, we have performed reductive titrations on the manganese-loaded forms of both of our wild type SODs, as well as the double mutant forms in which the metal specificity had been observed to be switched. Each protein was fully oxidized with excess permanganate followed by dithionite titration, with the abundance of the oxidized form, Mn(III), being measured by UV/visible absorption spectroscopy. These data demonstrate that the camSOD is reduced by lower concentrations of dithionite than the MnSOD, whereas this trend is reversed in the double mutant forms. These data are consistent with our proposed redox tuning mechanism, illustrate that the reduction potentials differ between MnSOD and camSOD, and that these differences are caused by the residues at sequence positions 159 and 160. These data complement our existing EPR and auto-oxidation data.

We would note that, since their discovery, measuring reduction potentials of SOD enzymes has been and continues to be extremely technically challenging. The redox-active metal ion in this family of enzymes is deeply buried within the protein fold. Access to the metal cofactor is limited to the narrow solvent channel that is proposed to enable substrate/product diffusion to/from the active site metal. This channel can be narrow because of the small size of the substrate (O_2^-) and products (O_2 and H_2O_2). Together, these factors prevent efficient electron transfer between the metal cofactor and the electrodes that would be required to measure the potential by standard electrochemical methods such as cyclic voltammetry. In a very few cases, some experimental success has been achieved using redox-active mediator molecules with SODs, but more often than not they have not been successful (Lawrence & Sawyer, 1979, Whittaker & Whittaker, 1991, Verhagen *et al.*, 1995, Vance & Miller, 1998, Leveque *et al.*, 2001, Li *et al.*, 2014). The mediator molecules that are generally used are significantly larger than these substrates/products, and thus also have difficulty interacting with the SOD metal cofactor. In addition, SOD enzymes (including the staphylococcal SODs) often show evidence of significant enzyme degradation when they go through oxidation/reduction cycles *in vitro* (Lawrence & Sawyer, 1979, Whittaker & Whittaker, 1991, Verhagen *et al.*, 1995, Leveque *et al.*, 2001). For these reasons, and although many researchers and laboratories have tried, there are remarkably few reports in the literature in which true midpoint reduction potentials have been unambiguously measured at equilibrium for any Mn/Fe-dependent SOD isozyme from any biological system. Where quantitative numbers have been presented, these are estimated values that are based on a unidirectional titration, e.g. data acquired from a reductive titration but not complemented by an oxidative titration due to protein instability. Despite this limitation, the concept of redox tuning of the SOD metal cofactor by the protein architecture is well established (Grove *et al.*, 2008, Miller, 2008, Sheng *et al.*, 2014, Vance & Miller, 1998), and therefore our suggestion that the likely mechanism by which our mutations affect metal specificity in the staphylococcal SODs is consistent with the body of existing literature. In the re-written text of the Results describing the spectroscopic evidence for differences in reduction potential, we have included a sentence that makes this point for the reader, which states "The SOD metal site is only accessible via a narrow solvent channel, making small redox-active

molecules inefficient electrochemical mediators for SODs, including the staphylococcal SODs (data not shown), thus precluding direct reduction potential measurements” (lines 237-40, page 10).

During this revision, we have attempted to utilize the method of Silverman (Leveque *et al.*, 2001) to estimate the reduction potentials of these SODs. We found that ferricyanide does appear to very weakly mediate between our staphylococcal SODs and an electrode, but that this process is extremely inefficient and is kinetically very slow, taking several days to reach equilibrium, over which period it is very challenging to maintain anaerobic conditions within the spectrophotometer and protein stability interferes with measurements. This has hampered our attempts to measure the reduction potential of the staphylococcal SODs using the Silverman methodology, despite significant efforts. We have prepared and used >300 mg of selectively manganese-loaded recombinant SOD enzymes for the experiments during the revision period, significantly more than was used to collect all the previous data in the original manuscript, and expended eight person-months of effort. Despite this, we have been unable to determine a true midpoint reduction potential. Importantly, the Silverman group noted that the method that they employed with limited success on the human MnSOD enzyme was incompatible with determining the reduction potential of the *E. coli* SODs (Leveque *et al.*, 2001). Consistent with studies using other SODs, it suggests a significant technical hurdle must be overcome before true midpoint reduction potentials can be assessed for these enzymes. Once those barriers are overcome, though, we feel that the staphylococcal SODs will be a unique model system with which to study the molecular mechanisms by which redox tuning is regulated by the protein architecture. This was previously implied in the Discussion (now on lines 394-5, page 15) when we stated that our data “highlight the potential of the staphylococcal SODs as a model system with which to uncover the mechanisms by which redox tuning is controlled”.

In the reviewer's opinion, the manuscript does not fit the broader readership of Nature Communication and would be more suitable to a specialized journal.

13. We respectfully disagree with the reviewer's opinion that our work would be more appropriate for a specialized journal. Similar to the other reviewers, who stated, “*The work is well designed and the results are a valuable contribution to research in the field of bioinorganic chemistry and beyond. I thus expect that this paper will obtain broad interest throughout the scientific community*” and “*The ability to examine the evolution of the two SODs and say something about metal use and metal stress is remarkable. To introduce mutations into an enzyme and alter the specificity of metal utilization is also impressive*”, we believe that our work substantially advances our understanding of an important area of science and will appeal to a broad audience, including investigators interested in metalloprotein function, SOD structure and function, protein and microbial evolution, and microbial pathogenesis. We would further note that the reviewer states that “*Determining the factors that can alter metal cofactor reactivity are important for both medicine and biotechnological applications. Overall the work is robust and well conducted*”, which speaks to broad applicability and potential interest in our results.

Minor comments:

The authors refer to Mn(II) oxidation state in line 197, but it sounds more to electronic structure or redox potential because as it is Mn(II) the oxidation state cannot be modified unless it is not Mn(II) any more. Please clarify.

14. This section of the document has now been re-written (see points 12 and 16), and this sentence has been omitted as a result.

The authors often refer to metal specificity. The reviewer would suggest metal reactivity because metal specificity would better correspond to either Mn or Fe preferentially bound to the active site. Please clarify this point.

15. We respectfully disagree with the reviewer on this issue. The terms 'metal selectivity' and 'metal specificity' are already in widespread usage for the phenomena that they describe. 'Metal specificity' is the ability of a metalloprotein to use a given metal, while 'metal selectivity' refers to the ability of a protein to select (or preferentially bind) the correct metal ion. Importantly, while metalloenzymes can be specific, due to the Irving-Williams series it seems that most metalloproteins are not selective (Dainty *et al.*, 2010, Tottey *et al.*, 2008, Waldron & Robinson, 2009). We did not invent these terms; indeed, a nice definition of these terms is given in the paper mentioned by reviewer 2, which states that: "Unfortunately there seems to be no simple explanation for the selectivity (during *de novo* protein folding) or specificity (enzyme activity with a selected metal) of this ubiquitous and important class of enzymes" (Hunter *et al.*, 2018).

Reviewer #4 (Remarks to the Author):

This paper aims to identify key structural and physical properties changes that allow SOD enzymes and mutants thereof from a bacterium to utilize two different ions, Mn and Fe. HF EPR is used to determine the zero-field interaction of the various mutants which is very sensitive to the electronic structure and coordination symmetry. The spectral differences are shown in Figure 3a which are small but certainly significant. The authors then site a paper where in model compounds these differences in zero-field interaction (D and E) are correlated with redox potential. As this is an enzyme and the paper aims to characterise structural and physical differences, the actually redox potentials should be measured to confirm this is actually true in this case.

16. See point 12, above, for a more thorough response to this issue of reduction potentials. Although we agree that measurements of midpoint reduction potentials would be valuable to understand the molecular mechanism of regulation of metal reactivity, we feel that the strength of our study lies in determining the evolutionary mechanism by which metal specificity has been shaped. We also feel that our auto-oxidation data, EPR analysis, and our new qualitative titration data are supportive of our interpretation that reduction potential determines reactivity, which in turn is consistent with a wealth of literature that describe the redox tuning model for SOD function.

This data could be obtained by an EPR redox titration, clearly the Authors demonstrate that the EPR signal is quite strong and so this type of EPR experiment should be feasible (at X-band or W-band) even if a bit tedious.

17. The limiting factor in determining accurate reduction potentials of the SOD enzymes lies not in measuring the abundance of Mn(II) or Mn(III) during the titration, which can be visualized by EPR and UV/visible spectroscopy respectively, but in measuring the electrochemical potential using electrodes. As described in the previous comment (see point 12), this is because the small molecules commonly used for such redox experiments with metalloproteins mediate between the metal cofactor and the electrode only very poorly in this family of enzymes, and equilibrium requires very long periods of incubation. Therefore, such experiments using EPR to quantitatively detect Mn(II) during a redox titration suffer from the same technical hurdles as` described above for the spectroelectrochemical approaches we have tested.

'The Authors state in the conclusions and elsewhere that the alter redox potential is a main cause of the change in reactivity, but it was never measured. The uncertainly in this model complex correlation to this enzyme is highlighted by Supplementary Figure 5 were D/E ratios do not fit the model compounds. The redox potentials needs to be measured for the paper to be impactful, as the Authors write in their conclusion on page 13, line 319: 'enabling future studies to determine how the architecture of a metalloenzyme imposes cofactor specificity'. Meaningful redox potentials would be very important for this future aim.

18. Supplementary Figure 5 does not show *D/E* or *E/D* ratios, which for other metals is a measure of coordination site symmetry. It is arguable whether *E/D* can be used for Mn(II) complexes in this way (Gätjens *et al.*, 2007, Tabares *et al.*, 2005). The x-axis of Supplementary Figure 5 is $|D|+E$, which a measure of the strength of the ZFI. There was no intent to make the comparison

quantitative. The absolute size of $|D|+E$ values are very different for the terpyridine complexes compared with the Mn-loaded SOD enzymes, but they follow trends that are analogous to those in the SODs. The similarities between the terpyridine model complexes and SODs is striking not only in terms of ZFI but also redox potential.

We agree that measurements of the reduction potentials would be valuable for delivering impact from our study of the staphylococcal SODs. However, as explained in point 12, above, performing such measurements on SOD enzymes is extremely difficult, and solving the technical issues associated with these measurements for the SODs will be an objective over the coming years for the team and our collaborators. We would also reiterate that this study has made a breakthrough discovery by uncovering the evolutionary mechanism by which altered metal specificity was achieved, rather than the molecular mechanism by which it was achieved.

Minor points

On page 7, line `63, the Authors state that the positions of the mutated residues are no altered. This is at first a confusing statement because they must mean the position of the backbone as the side-chains are clearly different. This sentence and the next ones (like page 7, line 176 & page 8, line 194) should be rewritten.

19. The reviewer is correct to point out that we meant the position of the polypeptide backbones were not altered, especially when referring to the structures of the mutated variants which by definition have different sidechains. This has been corrected on line 111 and line 182.

In this regard, the accuracy of the X-ray structure should be discussed, in particular the bond lengths and angles around the Mn ion (i.e. describe briefly Supplementary Table 4 and the errors).

20. We have included a comment specifically referring to the “insignificant changes to metal-ligand bond lengths or angles within the crystallographic resolution” on lines 113-4 on page 5. We have added a row to Supplementary Table 4, which contained the bond distances, that includes the resolution of the crystal structures to ensure the reader is aware of the limits to which they should interpret these data. Given the limited resolution of some structures, it is important that the precision of these bond distances are not over-stated. We have also added a new table, Supplementary Table 5, which gives the bond angles from the structures, as requested.

Page 8, line 186. ‘zero field interaction’, zero-field is one word and needs the hyphen

21. This error has been corrected in the main text of the manuscript – see line 192 on page 8 – and also in the supporting information – see line 100 on page 6 of the SI.

Figure 3A and tables Supplementary 5: The HF EPR spectra were analyzed using second-order expressions, which should be accurate enough to extract the D and E values (Supplementary Table 5). However, it would be beneficial to the reader to include a simulation of each spectrum using a full matrix diagonalization approach. This is a very easy task today and numerous free software packages are available to achieve this. These spectra should appear in the Supporting Material.

22. We have previously shown that the second perturbation equations are as accurate as matrix diagonalization methods and are more than adequate to determine accurately the zero-field D and E values (Tabares *et al.*, 2005, Un *et al.*, 2004). As discussed in those prior publications, full matrix diagonalization has no intrinsic advantage in accuracy. Neither the computational effort of the calculations nor the availability of software were factors in adopting our approach. Indeed, as discussed in the Online Methods, the inflection method used to determine the “edge” in conjunction with the perturbation equations has the significant benefit that no explicit model of the distributions in E and D is needed. This was important since it was evident that the shapes of the relevant features were not the same for each protein, most likely due to distributions in D and E . Equally important was the presence of systematic errors in measurement of the magnetic-fields. Unlike

conventional EPR, accurate field calibration and linear sweeps over 2 T is not possible with the magnet that was used in the EPR measurements. The field positions of the edges to first order (as indicated in Supplementary Figure 6) could have been “read off”. Rather than simply use these values, differences were used to reduce errors in field calibration and nonlinearities in sweep. This iterative, self-consistent approach yields *D* and *E* values to better than 10 MHz (Barwinska-Sendra *et al.*, 2018). Error analysis using our perturbation approach was simpler, since the method only required determining field position. Most matrix diagonalization methods, to the best of our knowledge, introduce simple distribution models for *D* and *E* in an *ad hoc* manner making accurate simulations difficult. Matrix and full simulation approaches require manual or iterative non-linear fitting that depend on minimizing root mean square the difference between measured and calculated amplitudes. This requires both accurate positions and a sufficiently accurate model of the line shape. Assessing errors is much more difficult. For such approaches, incorporation of systematic errors in magnetic-field is also not obvious. For these reasons, we have decided not to include any full simulations which, at best, would only provide visual guidance without imparting any quantitative information.

23. In addition to these changes in response to reviewers’ comments, a number of minor typographical errors have also been identified and corrected. Some further editing has also been undertaken to keep the manuscript within the desired length constraints and to meet the requisite formatting guidelines for *Nature Communications*.

References

- Andreini, C., Bertini, I., Cavallaro, G., Holliday, G.L., and Thornton, J.M. (2008) Metal ions in biological catalysis: from enzyme databases to general principles. *Journal of biological inorganic chemistry : JBIC : a publication of the Society of Biological Inorganic Chemistry* **13**: 1205-1218.
- Barwinska-Sendra, A., Basle, A., Waldron, K.J., and Un, S. (2018) A charge polarization model for the metal-specific activity of superoxide dismutases. *Physical chemistry chemical physics : PCCP* **20**: 2363-2372.
- Dainty, S.J., Patterson, C.J., Waldron, K.J., and Robinson, N.J. (2010) Interaction between cyanobacterial copper chaperone Atx1 and zinc homeostasis. *Journal of biological inorganic chemistry : JBIC : a publication of the Society of Biological Inorganic Chemistry* **15**: 77-85.
- Dupont, C.L., Butcher, A., Valas, R.E., Bourne, P.E., and Caetano-Anolles, G. (2010) History of biological metal utilization inferred through phylogenomic analysis of protein structures. *Proceedings of the National Academy of Sciences of the United States of America* **107**: 10567-10572.
- Garcia, Y.M., Barwinska-Sendra, A., Tarrant, E., Skaar, E.P., Waldron, K.J., and Kehl-Fie, T.E. (2017) A Superoxide Dismutase Capable of Functioning with Iron or Manganese Promotes the Resistance of *Staphylococcus aureus* to Calprotectin and Nutritional Immunity. *PLoS pathogens* **13**: e1006125.
- Gätjens, J., Sjödin, M., Pecoraro, V.L., and Un, S. (2007) The Relationship between the Manganese(II) Zero-Field Interaction and Mn(II)/Mn(III) Redox Potential of Mn(4'-X-terpy)₂ Complexes. *Journal of the American Chemical Society* **129**: 13825-13827.
- Grove, L.E., Xie, J., Yikilmaz, E., Miller, A.F., and Brunold, T.C. (2008) Spectroscopic and computational investigation of second-sphere contributions to redox tuning in *Escherichia coli* iron superoxide dismutase. *Inorganic chemistry* **47**: 3978-3992.
- Hsieh, Y., Guan, Y., Tu, C., Bratt, P.J., Angerhofer, A., Lepock, J.R., Hickey, M.J., Tainer, J.A., Nick, H.S., and Silverman, D.N. (1998) Probing the active site of human manganese superoxide dismutase: the role of glutamine 143. *Biochemistry* **37**: 4731-4739.
- Hunter, T., Bonetta, R., Sacco, A., Vella, M., Sultana, P.-M., Trinh, C.H., Fadia, H.B.R., Borowski, T., Garcia-Fandiño, R., Stockner, T., and Hunter, G.J. (2018) A Single Mutation is Sufficient to Modify the Metal Selectivity and Specificity of a Eukaryotic Manganese Superoxide Dismutase to Encompass Iron. *Chemistry – A European Journal* **24**: 5303-5308.
- Kirschvink, J.L., Gaidos, E.J., Bertani, L.E., Beukes, N.J., Gutzmer, J., Maepa, L.N., and Steinberger, R.E. (2000) Paleoproterozoic snowball earth: extreme climatic and geochemical

global change and its biological consequences. *Proceedings of the National Academy of Sciences of the United States of America* **97**: 1400-1405.

Lawrence, G.D., and Sawyer, D.T. (1979) Potentiometric titrations and oxidation-reduction potentials of manganese and copper-zinc superoxide dismutases. *Biochemistry* **18**: 3045-3050.

Leveque, V.J., Vance, C.K., Nick, H.S., and Silverman, D.N. (2001) Redox properties of human manganese superoxide dismutase and active-site mutants. *Biochemistry* **40**: 10586-10591.

Li, W., Wang, H., Chen, Z., Ye, Q., Tian, Y., Xu, X., Huang, Z., Li, P., and Tan, X. (2014) Probing the metal specificity mechanism of superoxide dismutase from human pathogen *Clostridium difficile*. *Chemical communications (Cambridge, England)* **50**: 584-586.

Miller, A.F. (2008) Redox tuning over almost 1 V in a structurally conserved active site: lessons from Fe-containing superoxide dismutase. *Accounts of chemical research* **41**: 501-510.

Sheng, Y., Abreu, I.A., Cabelli, D.E., Maroney, M.J., Miller, A.F., Teixeira, M., and Valentine, J.S. (2014) Superoxide dismutases and superoxide reductases. *Chemical reviews* **114**: 3854-3918.

Sjodin, M., Gatjens, J., Tabares, L.C., Thuery, P., Pecoraro, V.L., and Un, S. (2008) Tuning the redox properties of manganese(II) and its implications to the electrochemistry of manganese and iron superoxide dismutases. *Inorganic chemistry* **47**: 2897-2908.

Tabares, L.C., Cortez, N., Agalidis, I., and Un, S. (2005) Temperature-Dependent Coordination in *E. coli* Manganese Superoxide Dismutase. *Journal of the American Chemical Society* **127**: 6039-6047.

Totter, S., Waldron, K.J., Firbank, S.J., Reale, B., Bessant, C., Sato, K., Cheek, T.R., Gray, J., Banfield, M.J., Dennison, C., and Robinson, N.J. (2008) Protein-folding location can regulate manganese-binding versus copper- or zinc-binding. *Nature* **455**: 1138-1142.

Uberto, R., and Moomaw, E.W. (2013) Protein similarity networks reveal relationships among sequence, structure, and function within the Cupin superfamily. *PloS one* **8**: e74477.

Un, S., Tabares, L.C., Cortez, N., Hiraoka, B.Y., and Yamakura, F. (2004) Manganese(II) Zero-Field Interaction in Cambialistic and Manganese Superoxide Dismutases and Its Relationship to the Structure of the Metal Binding Site. *Journal of the American Chemical Society* **126**: 2720-2726.

Valasatava, Y., Rosato, A., Furnham, N., Thornton, J.M., and Andreini, C. (2018) To what extent do structural changes in catalytic metal sites affect enzyme function? *Journal of inorganic biochemistry* **179**: 40-53.

Vance, C.K., and Miller, A.F. (1998) A Simple Proposal That Can Explain the Inactivity of Metal-Substituted Superoxide Dismutases. *Journal of the American Chemical Society* **120**: 461-467.

Vance, C.K., and Miller, A.F. (2001) Novel insights into the basis for *Escherichia coli* superoxide dismutase's metal ion specificity from Mn-substituted FeSOD and its very high E(m). *Biochemistry* **40**: 13079-13087.

Verhagen, M.F.J.M., Meussen, E.T.M., and Hagen, W.R. (1995) On the reduction potentials of Fe and Cu□Zn containing superoxide dismutases. *Biochimica et Biophysica Acta (BBA) - General Subjects* **1244**: 99-103.

Waldron, K.J., and Robinson, N.J. (2009) How do bacterial cells ensure that metalloproteins get the correct metal? *Nature reviews. Microbiology* **7**: 25-35.

Whittaker, J.W., and Whittaker, M.M. (1991) Active site spectral studies on manganese superoxide dismutase. *Journal of the American Chemical Society* **113**: 5528-5540.

REVIEWERS' COMMENTS:

Reviewer #3 (Remarks to the Author):

Anna Barwinska-Sendra and co-workers present a corrected version of their previous manuscript entitled "An evolutionary path to altered cofactor specificity in a metalloenzyme". They have, in the current version, addressed all the reviewer's requests and I have no further remark or question. This manuscript is a very neat study and should deserve publication.

Reviewer #4 (Remarks to the Author):

Specific Comments to Address

Point 16. The authors have tried to measure the redox potential and thoroughly demonstrated the difficulty in obtaining such measurements. Text was included now in the main article to explain to the reader why this important parameter is missing. A new supplementary Figure 8 was added to help address the problem. I am satisfied with their response to this question.

Point 17. This is similar to point 16, see my comment above.

Point 18: Given that the redox potentials cannot be measured, Figure 5 provides at least proof that the electronic environment is significantly altered. At this stage, without the redox potentials, this is all that can be said so Figure S5 provides indirect evidence and thus is worthy of inclusion. I therefore accept the author's response. This is now also supplemented by a new Figure 8 (Supplementary) which also trends in the correct direction.

Point 19: This was fixed, satisfied

Point 20: This was addressed and data are provided in Supplementary Table 5.

Point 21: fixed

Point 22: I accept the response given and their conclusions, it is also backed up by references so I am happy with the second order perturbation equation approach for the purpose they are using it for.

The points raised in my review (some of which were raised by the other reviewers) have been addressed and where data is difficult to obtain it is now adequately discussed, so the reader can better appreciate the work. The main point of the paper, showing different metal specificity of the

two enzymes and how this is influenced by reciprocal amino acid substations is robust work and will be of interest in my opinion to the readers of Nature Communications. I therefore now recommend this article for publication.